

# 1 Shift of seed mass and fruit type spectra along longitudinal gradient: high

# 2 water availability and growth allometry

Shunli Yu[1],Guoxun Wang[1],Ofir Katz[2],Danfeng Li[1],Qibing Wang[1],Ming Yue[3], Canran Liu[4]
[1] State Key Laboratory of Vegetation and Environmental Change, Institute of Botany, Chinese Academy of Sciences
[2] Dead Sea and Arava Science Center, Israel
[3] College of Life Sciences, Northwest University, China
[4] Rylah Institute for Environmental Research, Heidelberg, Department of Environment, Land, Water and Planning,
Melbourne, VIC 3084, Australia.
Correspondence to: Shunli Yu (shunliyu@ibcas.ac.cn)
**Abstract.** Propagule traits vary among communities along geographical gradients such as longitude, but the
mechanisms that underlie these variations remain unclear. This study aims to explore seed mass variation patterns of
different community types along a longitudinal gradient and their underlying variation mechanisms by involving an
in-depth analysis on the variation of seed mass, fruit type spectra, growth forms and dispersal mode spectra in Inner
Mongolia and northeastern China. Plant community characterization and seed collection were conducted in 26 sites
spreading over five vegetation types and covering 622 species belonging to 66 families and 298 genera. We found
there are significantly declining trend for mean seed mass, vertebrate-dispersed species richness and fleshy-fruited
species richness along a longitudinal gradient from forests to desert grasslands. However, we also found the lowest
average seed mass and the smallest proportion of species dispersed by vertebrates occurring at typical grasslands in the
five communities. The variations of average seed mass display high congruent with transition of growth forms. The
selection for these propagule attributes is driven mainly by climatic factors such as precipitation, temperature, soil
moisture and evaporation, as well as by internal biotic factors such as growth forms, canopy coverage and leaf area. A
hypothesis was provided that environmental factors and botanical traits that favor greater water availability lead to
emergence (or speciation) of species with large seeds or fleshy fruits with high water content. Due to greater water





availability and increasing leaf area, much more photosynthate and allometric growth then ultimately increase the
community average seed mass along rising longitude (or declining latitude or elevation). Phylogenetic signal or
diversity are not found to be involved into the effect on the patterns. A novel mechanistic framework and model are
provided to expound seed variation among species or communities.

## 1 Introduction

Seed traits have great impact on plants' competitive ability, fitness, fecundity and reproduction. Therefore, studying
seed mass variations and their mechanism is crucial for understanding plant's ecological strategy and resource
acquisition (Zanne et al., 2014) as well as deep-time origin and evolution of seed attributes. Being a relatively stable
biological trait, seed mass is controlled by an appointed gene (Adamskia et al., 2009; Wang et al., 2014), while
retaining a certain degree of plasticity being affected by the surrounding environment (Baker, 1972). Therefore, an
individual plant's seed size is a combined result of its taxonomic group's evolutionary history and of immediate
selective pressures of the local environment (Westoby et al., 2002; Moles et al., 2005a). Furthermore, as an important
aspect in the reproductive biology of plants, seed mass is evolutionarily associated with and corresponds to other plant
traits, relating to plant habits, life history such as growth forms (Moles et al., 2005a), stature and canopy sizes (Venable,
1992; Leishman and Westoby, 1994; Moles et al., 2005a), dispersal modes (Greene and Johnson, 1993), leaf area (Díaz
et al., 2016) and plant longevity (Telenius and Torstesson, 1991), as well as to anatomical traits of flowers and fruits
(Primack, 1987).
Numerous works show that seed mass varies along environmental gradients such as latitude, elevation and
longitude owing to environmental variations in temperature and precipitation and canopy coverage both among and
within communities (Moles et al., 2007) and several mechanisms are proposed to explain such seed mass variation
gradients or patterns, for example, temperature (Moles et al., 2014), light (or solar radiation) (Murray et al., 2004;
Demalach and Kadmon, 2018), soil resource availability (Demalach and Kadmon, 2018), growth forms (Moles et al.,
2005b), dispersal modes (Moles and Westoby, 2003), soil pH (Tautenhahn et al., 2008) etc. However, a deep
understanding of the factors that underlie these major biogeographical variations is missing (Demalach and Kadmon,



2018), especially at a continental scale along longitude. Previous work suggested that community-level average seed mass tends to decrease towards higher latitudes and elevations (Moles et al., 2007). These trends can be explained by shifts in habitat type, plant growth form spectra, seed disperser assemblage (Moles and Westoby, 2003), solar radiation and metabolic expenditure (Murray et al., 2003; 2004) and NPP (Bu et al., 2007; Guo et al., 2010) along latitudinal and elevational gradients. Additionally, species that prefer shaded habitats and late successional stages generally tend to have larger seeds than those in open arid habitats or earlier successional stages (Baker, 1972; Salisbury, 1974; Foster and Janson, 1985; Hallett et al., 2011; Moles and Westoby, 2006), indicating a strong effect of high water availibility on seed mass owing to low evaporation under close canopy coverage. Longitudinal variations of seed mass has been discussed among species with a single genus (Murray et al., 2003; 2004); however, there are few studies that focus on how community-level variations of seed mass (especially across species) correspond with other plant traits along longitudinal gradients, because of the difficulty to predict variations of comprehensive environmental factors arising from complex topography. Average seed mass is expected to decrease with declining longitude due to gradually less rainfall from forests to desert ecosystems (Murray et al., 2003; 2004). Here we present a study of community-level variations in seed mass in correspondence to position in the continent (relative to the sea) across Inner Mongolia and northeastern China, to identify the longitudinal pattern and discuss the mechanisms that may underlie them.

Previous works emphasize the role of high light acquisition and allometric growth in shaping seed mass variation through model prediction and experiment testing (Demalach and Kadmon, 2018; Demalach et al., 2019), and in this article we emphasize the importance of high water availability and allometric growth for speciation and colonization of species with large seeds due to environmental factors and biological traits. Allometry of biomass growth and size-asymmetry of light competition became the drivers of seed mass variation owing to soil resource availability and ultimate productivity heterogeneity along soil resource gradient (surely including water gradient). As we know, primary production of communities increases across an increasing water gradient (Bai et al., 2008). This article presents a novel mechanistic framework that integrates previous theory and hypotheses (related to climate, phylogeny, water conduction systems and other traits related to water balance) to evaluate seed mass variation among species or communities (Figure 1).





The objectives of this study are to explore seed mass variation patterns of different community types along a
longitudinal gradient and seed mass variation mechanisms. First, we test whether community-level seed mass declines
from forests to deserts and what drives the patterns by identifying correlations between average seed mass and
precipitation and temperature. Second, we test whether species richness and growth form display similar variations as
seed mass, because growth forms are the key determinant of seed mass (Moles et al., 2005b). Third, we test whether
seed mass variations are significantly associated with growth forms, fruit types and dispersal modes in order to know
whether woody species combine larger seed or fleshy-fruited species. Fourth, we expound whether these patterns can
be simply explained by phylogeny, latitude and elevation. Finally, we construct a general hypothesis for seed mass
evolution based on our results and conclusions. In addition, like most plant functional traits, seed size and fruit water
content are also considered to be related to phylogenetic history (Griffiths and Lawes, 2006; Norden et al., 2012).
These traits are often strongly phylogenetically conserved, so phylogenetic distances need to be considered when
examining trait variation patterns and their correspondence to environmental variables (Griffiths and Lawes, 2006; Yu
et al., 2017). Therefore, we also took into account the effect of phylogenetic signal or diversity on the seed mass
distribution patterns in our analysis.

**2 Methods**
**2.1 Study sites and plant community characterization**
The study area is located in continental arid Inner Mongolia plateau, where vegetation types shift from broad-leaved
deciduous forest to typical grassland and finally to desert (from east to west, respectively), due to a gradual increase in
sunshine duration and intensity and decrease in rainfall (from 780.6 to 29 mm) (Table 1). The study was carried out in
26 sites along this gradient, extending between longitude 100°E ~ 124°E (about 2864 km distance between the
westernmost and easternmost sites) and 41°N ~ 44° N in Inner Mongolia and northeastern China (Table 1).
Different sampling designs were used in different habitat types, owing to differences in vegetation structure and
density. Within each forest plot, 6 quadrats of 10×10 m²  were selected at random in undisturbed or slightly disturbed
(at least in recent years) areas. For shrub communities and herbaceous communities, 3 quadrats of 5×5 m² and at least



8 quadrats 1×1 m² were investigated, respectively. Species composition (species number and number of individuals per
species) were recorded. The geographical positions (latitude, longitude and elevation) were measured by wireless GPS
logger (HOLUX Technology Inc., Taiwan). Other physiographic factors such as grade of slope, slope aspect and
micro-topography were recorded. The vegetation types were determined based on the dominant species and
information referenced from the classification system of Chinese vegetation (Editorial Committee for Vegetation of
China, 1980). Data of temperature and precipitation as well as other climatic factors were retrieved from the Wordclim
database using R raster package: average values per site were obtained from interpolations of observed data during the
period between 1950 and 2000. Two climatic variables, mean annual temperature (MAT) and mean annual
precipitation (MAP) were used to analyze the relationship between seed mass and the climate along the geographical
gradient (Table 1).

**2.2 Seed collection and characterization**
Mature seeds were collected for each species observed in each site at the start of natural dispersal season (from June to
October) during the years 2008-2014. Seeds of each species were collected from at least three mother individuals in
the same stand and mixed together to avoid bias caused by maternal effects. Seeds were allowed to air-dry to a
constant mass in the laboratory before being weighed. For each plant species, seed mass was calculated as the average
values of 5 to 100 (even to 1000) seeds, depending on seed size and availability, and at least three repeated
measurements were conducted for each species. Seed mass was measured to microgram precision on a PB303 balance
(Mettler, Toledo). Seeds that were likely to be inviable (unusually small seeds that contained abnormal looking
embryos or that appeared hollow) were subjectively excluded before the measurement. For some species with
caryopsis, achene and utricle, dry mass of entire propagules was weighed. Structures such as delicate wings and
pappus (or hairs) strictly associated with wind dispersal were removed and the spine was retained before weighing the
seeds.

Dispersal modes were assigned for all 622 species, based on ornamentation and appendages on fruits and seeds.

The dispersal modes of each species were obtained from the Kew Gardens (Howe and Smallwood, 1982) and literature



reference collection from northwest China (Liu et al., 2014). For some species, dispersal modes were confirmed with
empirical analysis according to morphological features of their diaspores, and the dispersal mode represents seeds from
the parent plant to the soil surface. Each species was treated as having a single dispersal mode, reflecting their
principal dispersal agents (PDA) (Leishman and Westoby 1994; Butler et al., 2007): wind-adapted (279 species, with
wings, hairs or a pappus), vertebrate-adapted (66 species, with an aril or fleshy fruits), ant-adapted (195 species, with
an elaiosome), unassisted (70 species, no obvious morphological structure) and adhesion-adapted (12 species, with
hooks, spines or bards).

**2.3 Fruit types and other plant traits**
Fruits were classified as fleshy if they were described in the flora as berries, drupes, pomes, rose hips, multiple fruits
and pepos or as possessing fleshy pericarp or succulent tissue in general (including arils) (Yu et al., 2017). Accordingly,
capsules, achenes, nuts, caryopses, legumes, follicles, pods, cremocarps, utricles, samaras and schizocarps were
classified as non-fleshy (dry) fruits. Some species (e.g., *Vitex nengudo* var. *heterophylla*) that were described to be
fleshy-fruited in related local flora were reclassified as dry-fruited owing to very low water content.
Species life-history information is drawn primarily from the Flora of China and based on our yearly field
observations. Species in the flora were grouped into the following five major growth-form categories: trees (12
species), shrubs (65 species), subshrubs (20 species), lianas (15 species), perennials (400 species) and annuals (110
species).

**2.4 Soil moisture measurement**
The soil moisture of top 10-cm depth was measured gravimetrically by oven-drying the samples at $105^{\circ}$ C for 24 hours
in 12 sites of typical grasslands and desert grasslands. Five soil samples were collected from each sites on July 10-17,
2014, 21days after rainfall.

**2.5 Data analysis**



Seed mass, longitude and precipitation were log-transformed before analysis to meet the normality and
homoscedasticity assumptions of linear regression models. In order to ensure that any observed seed mass variation
along the longitudinal gradient is independent of latitude and elevation, general linear models (GLM) were employed.
Seed mass and other plant traits were treated as the dependent variable in all analyses with latitude, longitude and
climatic variables entered into models as independent variables.
The proportions and species richness of plants with various seed mass and fruit types in different communities
were compared using analysis of variance (ANOVA). ANOVA was also used to compare average seed mass between
different growth forms, different community types, different fruit types and dispersal types. The GLM procedure was
used to examine the explanatory power of community types, dispersal types, longitude, precipitation and temperature
on seed mass. All analyses were performed with R-3.3.3 (R Core Team, 2018). By use of the function commonality in
the R package yhat (Nimon et al., 2013), we take the log-transformed seed size as dependent variable, life forms,
vegetation types, dispersal modes and latitude as independent variables, exploring predictive power of each variables.
For the 620 species (two gymnosperm species were excluded owing to their low relatedness with most of
angiosperm species), a supertree was constructed using the software Phylomatic (Webb et al., 2008). The phylogenetic
backbone was based on the APG III tree (R20120829, http://phylodiversity.net/phylomatic/). We quantified the
strength of phylogenetic conservatism and tested the phylogenetic signal in seed mass using Pagel's $\lambda$ (Pagel, 1999)
and Blomberg's $K$ (Blomberg et al., 2003) calculated using the 'phylosig' function in the package 'phytools' v0.2-1 (R
Foundation, Vienna, Austria) (Revell, 2012). A $\lambda$ or $K$ of 0 indicates no phylogenetic signal (Pagel, 1999; Panchen et
al., 2015). Regression analyses were conducted between phylogenetic signal and longitude across the sites in five
community types. Using the phylogenetic tree with branch length, we calculated the phylogenetic diversity using the
measure PD, which was defined as the minimum total length of all the phylogenetic branches required to span a given
set of taxa on the phylogenetic tree (Faith, 1992). Taking mean seed mass as dependent variable and longitude
(including both linear and quadratic terms) and the phylogenetic diversity measure PD as independent variables, we
built a linear model using R package stats (R Core Team, 2018).
We considered the relations between the number of species with fleshy fruits and longitude, the number of





families, number of genera, and the phylogenetic diversity PD. Since there are strong correlations between the latter
four variables (r > 0.67, p < 0.001), they cannot be used in the same model. Therefore, we built four models. Each took
one of the four variables as the independent variable and the number of species with fleshy fruits as dependent variable.
A generalized linear regression model with Poisson family was fitted using R package stats (R Core Team, 2017). In
the model, we also included log (number of species) as offset.

**3 Results**
**3.1 Seed mass variations along the longitudinal gradient**
Although the majority of species had medium-sized seeds (Figure 1), variations among all species were great. There
were considerable differences in average seed mass and seed spectra among the five community types (Figure 2).
Forests have the largest average seed mass (23.45±18.34 mg) and both typical grasslands (4.75±3.93 mg) and sparse
forests (4.45±1.18 mg) have the lowest average seed mass, being nearly 5-fold decline in average seed mass from
forests to typical grasslands. Average seed mass of forests is significantly greater than that of the sparse forests
(F=12.13, $p$=0.0253), and deserts are remarkably larger (20.12±8.26 mg) than desert grasslands (10.08±2.34 mg)
(F=6.914, $p$=0.0466), being nearly 4.2-fold decline in average seed mass from deserts to typical grasslands. The
average seed mass of typical grasslands is significantly smaller than that of desert grasslands (F=11.92, $p$=0.0025),
while there is no significant difference between average seed mass of typical grasslands and sparse grasslands
(F=0.019, $p$=0.892).

**3.2 Seed mass relations to other plant traits**
Average seed mass of trees was significantly larger than that of shrubs (F=12.2, $p$=0.000), shrubs had larger seeds than
perennials (F=59.57, $p$=0.000), and average seed mass of perennials was larger than that of annuals (F=4.932,
$p$=0.0268) (Figure 3). From forests to grasslands, wooden species richness displayed a declining trend along
decreasing longitude (Table 2).
Seeds that are dispersed by vertebrates (232.09 ± 823.98mg) were significantly larger than those dispersed by





wind (2.46±6.23 mg) (F=238.2, $p$<0.0001), ants (3.56±10.03 mg) (F=17.73, $p$<0.0001), and those with unassisted

dispersal (7.42±12.08 mg) (F=17.73, $p$=0.000) and adhesive dispersal (5.07±8.12 mg) (F=17.73, $p$<0.0001) (Table 3).

Seed mass is weakly negatively correlated with leaf area ($R^2$ = 0.063, p = 0.005) and not significantly correlated

with SLA across all sampled species ($R^2$ = 0.006, p = 0.195). SLA is significantly related with leave area ($R^2$ = 0.160,

p < 0.001).

### 3.3 Seed mass relations to environmental variables

Average seed mass was minimum at approximately 112 degrees longitude where typical grasslands occur (Figure 4).

However, phylogenetic diversity (PD) was not a significant explanatory variable (p > 0.8) (Figure 4). Linear regression

model shows that there is no significant decreasing trend from forests to deserts along declining longitude (F = 2.289,

$p$ = 0.143). If the westernmost sample site (Ejinaqi) is excluded, seed mass significantly decrease inland ($R^2$ = 0.2434,

F = 7.398, $p$ = 0.012).

Significant negative relationships were found between seed mass and MAT ($R^2$ = 0.1752, $p$ = 0.01915) and

elevation ($R^2$ = 0.1221, $p$ = 0.0449) across all sample sites, but no significant relationships were found between seed

mass and latitude ($R^2$ = -0.028, $p$= 0.576) and MAP ($R^2$ = -0.008, $p$ = 0.380). Across 23 sample sites from desert

through desert grassland to typical grassland, average seed mass had significantly negative relationship with longitude

($R^2$ = 0.232, $p$ = 0.012) and MAP ($R^2$ = 0.48, $p$ = 0.00015), while across 20 sample sites from typical steppe to forests

average seed mass had significantly positive relationship with longitude ($R^2$ = 0.232, $p$ = 0.012) and MAP ($R^2$ = 0.48, $p$

= 0.00015). Average seed mass was found to just be weakly positive relationship with MAT both from desert through

desert grassland to typical grassland and from typical grassland to the forests ($R^2$ = 0.09207, $p$ = 0.08665). According

to above analysis, MAP should be crucial environmental drive factor for seed mass variation.

In addition, average seed mass is significantly related with soil moisture (p<0.05) and soil moisture significantly

decrease with declining longitude from typical to desert grasslands.

### 3.4 Species richness and proportion of fleshy fruited species



Average seed mass of species with fleshy fruits (40.15 ± 110.41 mg) were significantly greater than that with dry fruits
(26.58 ± 286.97 mg) (F = 18.61, $p$ = 0.0125) for the whole species pool (622 species), for the five community types
(Figure 3, SP 1) and for each sites (SP 2) (Figure 4).
Among the five community types, forests have the highest number (7.44 ± 1.26) and proportion (28.05 ± 6.16) of
fleshy fruited species, while desert grasslands have the lowest number (0.06 ± 0.097) and typical grasslands the lowest
proportion (1.00±1.49) (Figure 6). Fleshy fruited species richness (F = 22.25, $p$ = 0.00919) and proportion (F = 18.61,
$p$ = 0.0125) in sparse forests are significantly smaller than those in forests. The desert has higher fleshy fruited species
richness (F = 6.081, $p$ = 0.0239) and proportion (F = 24.9, $p$ < 0.0001) than desert grasslands. Sparse forests have
remarkably higher fleshy fruited species richness (F = 281.3, $p$= 0.000) and proportion (F = 78.6, $p$ = 0.0009) than
typical grasslands (Figure 6).

**3.5 Fleshy fruited species relations to environmental factors**
Fleshy fruited species richness was significantly associated with longitude ($R^2$ = 0.1691, $p$ = 0.02113) and MAP ($R^2$
= 0.4749, $p$ = 0.0000) across the 26 sample sites. Significantly positive correlation existed between the proportions of
species with fleshy fruits and MAT ($R^2$ = 0.1172, $p$ = 0.0486), while the correlation with elevation ($R^2$ = 0.0938, $p$ =
0.0703) and longitude ($R^2$ = 0.0831, $p$ = 0.0832) was weak. In addition, there were no significant relationships between
proportions of fleshy fruited species and latitude ($R^2$ = -0.0396, $p$ = 0.8272) as well as MAP ($R^2$ =- 0.0389, $p$ = 0.8009),
and no strong relationships between fleshy fruited species richness and latitude ($R^2$ = 0.0408, $p$ = 0.8899) as well as
MAT ($R^2$ = 0.0414, $p$ = 0.9416).
From desert through desert grassland to typical grassland, significantly positive correlations were found between
richness of fleshy fruited species and longitude (R=0.3466, $p$=0.0019) and MAP ($R^2$ = 0.284, $p$ = 0.0052), while there
were no significant correlations between proportion of fleshy fruited species and MAT ($R^2$ = 0.1295, $p$ = 0.0513). From
typical grassland to the forests, remarkable correlations occurred between proportion of fleshy fruited species and
longitude ($R^2$ = 0.324, $p$= 0.00418) and MAP ($R^2$ = 0.324, $p$= 0.00418), however no significant relationships were
found between MAT and proportion ($R^2$=-0.0519, $p$=0.9065) and species richness ($R^2$ = -0.0522, $p$ = 0.93) of fleshy





fruited plants. The number of species with fleshy fruits increased with longitude ($p = 0.022$) and number of families ($p$
$= 0.005$), but correlations with number of genera and phylogenetic diversity were not significant ($p = 0.056$ and $0.058$
respectively) (Figure 5, c-d).

**3.6 Phylogenetic signals and their shift along longitudinal gradient**
Most phylogenetic signals (k values) are weak (from 0.234 to 0.688, $p > 0.05$) for the five community types and for
most sample plots except Naimanqi (1.928, $p < 0.05$) in early-successional stage. No significant relationships are found
between phylogenetic signals (k values) and longitude across 26 sample sites (R = 0.0403, $p = 0.8596$). Both from
desert (to desert grasslands) to typical grasslands (R = 0.047, $p = 0.9123$) and from typical grasslands to the forests (R
= 0.0401, $p = 0.6382$), phylogenetic signals were not related to longitude for the five community types. Positive
relationships between longitude and number of families or species with fleshy fruits are significant ($p = 0.022$ and
0.005 respectively, Figure 5), and number of species and families with fleshy fruits increased as longitude increased.
However, relationships between number of genera or genetic diversity and longitude are not significant ($p = 0.056$ and
0.058), respectively (Figure 4).

**4 Discussion**
**4.1 Variation of seed mass spectra and environmental factors**
There is strong and consistent effect of community type (along a longitudinal gradient) on seed mass (Figure 2, Figure
4). The average seed mass display a significantly declining trend along decreasing longitude from forests to typical
grasslands and then to some sites in desert grasslands (Figure 4), however, mean seed masses increased from typical
grasslands to desert grasslands and desert ecosystems and then to forests (Figure 2), showing congruent distribution
patterns to plant growth forms (Table 2). MAT and MAP may be responsible for the results because of the significant
relationships between seed mass and MAT and MAP across 26 or 23 sample sites respectively (see results). This
indicates that climatic control of vegetation trait distribution in Inner Mongolia is not only by temperature but also by
precipitation (Moles et al., 2014). The combined effects of precipitation and temperature may be, to some extent, most



important to certain vegetation syndromes such as seed mass and fruit water content. High water availability
potentially can produce high assimilation products and high temperature (in normal range of plant growth) can increase
water availability.
General linear models (GLMs) revealed significant relationships between seed mass and each of the variables
predicted to influence the longitudinal gradient in seed mass: plant growth form (99.76%), vegetation types (99.01%)
and seed dispersal syndrome (99.88%). Such patterns have previously been attributed mostly to a correspondence of
seed mass to plant growth form and seed dispersal syndrome, which themselves are driven by climatic and
environmental variations (Moles et al. 2005a; Moles et al. 2007). In Inner Mongolia, typical grasslands are often
composed mainly of grasses (many of which are biennial and perennial) that are small-seeded (Figure 3), whereas trees
and lianas that dominate forests and shrubs that dominate deserts have the largest seeds (Figure 3). Large seeds were
proved to be often associated with woody growth forms (Salisbury, 1942; Baker, 1972; Silvertown, 1981; Mazer, 1989;
Jurado et al., 1991; Elenius and Torstensson 1991; Leishman and Westoby, 1994; Moles et al., 2005a; Moles et al.,
2005b). This pattern is often attributed to woody plants' better capability to take up (Schenk and Jackson, 2002; Li et
al., 2002; Qi et al., 2019) and store resources and to buffer effects of environmental variations on seed size (Weiner,
2004; Moles et al., 2005a), or to reduced evaporation for understory species (Yu et al., 2017). Surely, woody species,
on average, having larger leave, can produce more photosynthate to invest in seeds (Díaz et al., 2016).
It is possible that larger seeds are more common in drought-prone habitats most likely because they allow
seedlings to establish large root systems early, with a better chance of surviving drought (Baker, 1972; Salisbury, 1974).
In this study, desert grassland and desert ecosystems are found to be dominated by shrubs that often possess larger
seeds (Figure 3). These species are seldom exposed to strong interspecific competition or shading in Inner Mongolian
Plateau (Bai et al., 2008). In addition, relatively high species richness and the highest number of species occurred in
this typical steppe grassland (Table 2), and in contrast, desert steppe had very low species richness and number of
individuals (abundance) (Table 2).
With increasing MAP, richness of wind dispersed species decreased (S3, S4). The proportion of vertebrate
dispersed species in typical grasslands was the lowest in comparison to other communities (S4). The patterns of seed





dispersal syndromes observed in this study are congruent with previous findings in Australia's subtropics (Butler et al.,
2007). Biotic dispersal agents exert a strong selective pressure on angiosperm species with various seed size in Inner
Mongolian plateau, as evidenced by the evolution of a wide range of adaptations for animal dispersal.

**4.2  Variation of fruit type spectrum and associations of seed mass with fruit types**
Fleshy fruited species richness significantly corresponded to gradual changes of climate, especially for MAP (Table 1).
The smallest proportion of fleshy fruited species occurred in typical grasslands and desert grasslands (Figure 4), at the
middle zone of Inner Mongolia. Previous findings showed that fleshy fruited species were often associated with shaded
habitats, mature forests, tropical forests, regions with lower elevations and woody life form (summarized in Yu et al.,
2017), indicating high canopy coverage and low evaporation (Fig.6) (Yu et al., 2017). The increasing prevalence of
fleshy-fruited species with increasing canopy coverage (Table 2) is probably related to the prominence of species with
larger seeds in such habitats. Previous hypotheses suggest that fleshy fruit evolution is related to water availability and
the ineffectiveness of wind-assisted dispersal beneath a dense canopy (Butler et al., 2007; Yu et al., 2017). The reduced
dispersal capability following from an increase in seed mass may be counterbalanced by evolution of traits mediating
seed dispersal by animals, such as fleshy fruits. Alternatively, increasing water availability may promote the evolution
of species with fleshy fruits (Yu et al., 2017) and large seeds.

**4.3  Ecological and evolutionary drivers of seed mass variations**
A consistent combination was found between possession of fleshy fruits and heavier seeds when comparing seed mass
among clades with fleshy and non-fleshy fruits (Figure 3, S1, S2). The results were largely in agreement with previous
findings (Eriksson et al., 2000; Butler et al., 2007; Bolmgren and Eriksson, 2010). A possible explanation may be that
woody species have larger internal water surpluses and photosynthate to invest in their seeds and fruits. Accordingly,
formation of seed mass may also be related to plant resource acquisition and allocation strategies or to allometric
growth of plant apparatus (Weiner, 2004; Price et al., 2007; Demalach and Kadmon, 2018).

As a result of evolution for high water availability, large seeds are often associated with low latitude (Moles et al.,





2003) and low elevation (Bu et al., 2007), or with shaded habitats such as northern slopes (Csontos et al., 2004) and
closed vegetation (Mazer, 1989; Hammond and Brown, 1995) and with late successional stages (Hammond and Brown,
1995). All those phenomena indicate that seed mass may be related to low evaporation and high water availability in
plants (Fig.6). We suggest that, as an ecological strategy, the derivation and evolution of species with large seeds may
be due to improved water accommodation in plants by strong resource acquisition ability (such as having strong water
absorbing root system and advanced water conductive ability) or water retention ability (such as habituating shaded
environment or developing small, thick leaves and hair or waxiness on leaf to prevent water loss) (Baker, 1972;
Fonseca et al., 2000). Plant species have evolved various ecological strategies to match their environments (Laughlin,
2014). These strategies are manifested in many plant organs and traits. In the present study, seed mass is strongly
connected with other biological characteristics such as plant dispersal ability (SP 4, Table 3), fruit types and growth
forms. For example, there is rising trend in average leaf area (Wright et al., 2004) and water- conducting conduits
(Wheele et al., 2007; Zanne et al., 2014) along increasing longitude (or declining latitude and elevation). Seed mass
also is sure to be coordinated with conducting issues of plant apparatus (Wheele et al., 2007; Zanne et al., 2014).
Anatomical structures of lots of species indicated that the species with large seeds or fleshy fruits often have wide and
long vessel elements that can provide much more water (Carlquist, 1975; Zimmermann, 1983). As suggested before,
seed mass also is likely to be a result of co-evolution among various organs that determine plant responses to changing
abiotic factors (Díaz and Cabido, 1997; Sandel et al., 2010).

In light of growth allometry theory, average seeds mass variation should converge with community total biomass

(Demalach and Kadmon, 2018). Evidently, spatial distributions of community-level seed mass and NPP consistency
correspond (Moles and Westoby, 2003; Murray et al., 2003; Griffiths and Lawes, 2006; Chen et al., 2007; Chen et al.,
2011). In Inner Mongolia ANPP (aboveground net primary production) and RUE (rain-use efficiency) increased indeed
in space across different communities or ecosystems with increasing MAP eastwards (Bai et al., 2008), showing
similar variation trend with average seed mass. Both soil moisture and soil nutrient (total N) was found to decrease
significantly in Inner Mongolia from east to west (Liu et al., 2017), showing similar variation trend with ANPP and
seed mass. Moreover, water retention of plants is becoming unfavorable with increase of evaporation westwards (Table





1).

In previous studies, soil moisture was found to not correlate with the relative abundance of fleshy-fruited species

due to low temperature on water availability constraints (Yu et al., 2017). As we know, seed plants employ two main
strategies to increase water use efficiency: one is to take up more water through root systems and the other is reducing
water loss through evapotranspiration. In our study, canopy coverage decreases from forests to sparse forests and then
to grasslands and desert grasslands (Table 2), leads to gradual reduction in fleshy-fruited or large-seeded species
richness (Fig.1, Table 2). However, since fleshy fruits have high water content and thus inquire higher plant internal
water content (Yu et al., 2017), we suggest the correspondence of seed size and fruit water content imply that some
species evolved to contain more water or photosynthate in multiple body parts. Furthermore, $CO_2$ concentration is
generally the same everywhere although there is some small variation during growth seasons, its impact on seed mass
variation patterns should be expelled. Solar radiation is very similar along longitude especially among typical
grasslands, desert grasslands and deserts with similar elevation, its effect on seed mass variation should be partitioned
out. Therefore, combined with previous results of other studies, we deduce that drivers of seed mass spatial distribution
patterns include temperature, rainfall, solar radiation, soil moisture and nutrients, leaf area, canopy coverage and their
interactions, however, high water availability in plant body may be the most vital driving factor in shaping seed mass
spatial distribution. According to growth allometry, a fraction of photosynthate, coming from each increment of
temperature, rainfall, soil moisture and nutrients, leaf area, canopy coverage, is considered to be allocated to seeds. In
addition, biological structures (such as fair or waxiness on leaf to prevent water loss), that favor water retention in
plant body would also be useful in increasing seed mass or fruit water content.

Moreover, as suggested by previous works, transition between small and large seeds in response to environmental

variations should be genetically simple, involving suppression and re-expression of only a few genes (Wang et al.,
2014; Zanne et al., 2014). On basis of the above all, we suggested that transition between dry fruits and fleshy fruits in
response to environmental variations may also be genetically simple, involving suppression and re-expression of only a
few genes.

Generally, seed mass is quite phylogenetically conservative (Lord et al., 1995). However, in this study,





phylogenetic signal is weak across the 26 sites (Table 1) and the five communities and found to be little involved in the
relationships between seed mass and longitude, MAP and MAT. This proves that the environment affects seed mass in
the community context independent of phylogenetic constraints. The five communities are in middle or late
successional stages in which the main construction process is competitive exclusion rather than environmental filtering
(Norden et al., 2012).
A simple mechanistic model is provided to explain average (or total) seed mass variation between communities
for one species as following:
$$S_a = 1/n \sum_{i=1}^{n} C_{i1}B_t \ (C_{i1} < 1), \quad S_t = \sum_{i=1}^{n} C_{i1}B_t \ (C_{i1} < 1)$$
$$B_t = B_{id} + B_{i0} + B_l$$
$S_a$ is average seed mass of a community for one species (Fig.2), $S_t$ is the total seed mass of all species in a community,
n is number of species in a community, $C_{il}$ is the allometric growth coefficient (or allocation portion to seeds) that
differ among species. $B_t$ is total biomass from photosynthate per species. $B_{id}$ value is the biomass of photosynthate
related to water from conducting issues for one species, $B_{i0}$ is the biomass of photosynthate related to water from other
approach (for instances, lessening evaporation), $B_l$ is the biomass of photosynthate related to leaf area (Fig.1).
In additions, abundant groundwater in the desert ecosystem (Heihe river runs through Ejinaqi, the westernmost
sample plot in this study, here *Populus euphratica* sparse forest even develops around the river) may be responsible for
the "strange" patterns about distribution of average seed mass and fleshy fruited species found in this study. In this
study, we just measure the soil moisture of top 10cm which mainly influence growth of herbs, but for the growth of
shrubs and trees, rich soil water below the depth of 10 cm in some area of Ejinaqi also is useful. Moreover, ecological
scale and environmental heterogeneity often affects results of functional traits along biogeographical gradients, so
further study may be necessary in larger scale (or large area) to identify the results of this article.
**5 Conclusions**
Mean seed mass, seed dispersal spectra, fruit type spectra and plant growth forms of five community types vary
significantly along a longitudinal gradient, with the lowest average seed mass and the smallest proportion of species





dispersed by vertebrates occurring at the middle longitude (typical grasslands). The selection for these propagule
attributes is most likely to be driven by environmental factors such as precipitation, temperature, soil moisture, soil
nutrient and underground water, as well as many biological factors such as canopy coverage, plant morphologies and
life-spans, dispersal syndromes, densities of competing plants, and growth forms, however, water availability
potentials and growth-allometry may be key drivers of seed-mass variation along climatic gradients or resource
gradients. Larger seeded species may have evolved due to high water availability and more photosynthate. Our
findings have important implications in understanding origin and evolution of species with large seeds or fleshy fruits.
Further studies are needed to better understand the combined effects of climate, soil and evolutionary history on seed
mass variations among plants.

*Author contributions*
S.L. Yu led data collection and analysis, conceived the idea and led manuscript writing. G.X. Wang and D.F. Li took
part in data collection and analysis. O. Katz assisted in manuscript writing. C.R. Liu carried out phylogenetic analysis.
QB Wang provided soil moisture data. M. Yue gave a critical revision suggestions on early draft. All authors
contributed critically to the drafts and gave final approval for publication.

*Acknowledgements*
This study was funded by the National Natural Science Foundation of China (No. 40771070,41171041) and Beijing
Natural Science Foundation (No. 5092015).

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





**TABLE 1** Information geographic positions and environmental factors in 21 sampling sites in Inner Mongolia
plateau and Northeastern China

| Number | Site names | Longitude | Latitude | Altitude (m) | MAP(mm) | MAT (℃) | K-value | Evaporation (mm) | Vegetation types |
|---|---|---|---|---|---|---|---|---|---|
| 1 | Ejinaqi | 101.0815 | 41.9520 | 942 | 29 | 8.9 | 0.774 | 3850 | DS |
| 2 | Wulatehouqi | 107.0160 | 41.0750 | 1137 | 136.8 | 7.9 | 0.647 | 3069 | DG |
| 3 | Wulatezhongqi | 108.4833 | 41.3002 | 1203 | 198.8 | 5.8 | 0.395 | 2500 | DG |
| 4 | Erlianhaote | 112.0108 | 43.7239 | 907 | 134.8 | 4.6 | 0.451 | 2700 | DG |
| 5 | Suyouqi | 112.6328 | 42.7662 | 1100 | 346.8 | 2.1 | 0.855 | 2700 | DG |
| 6 | Chayouzhongqi | 112.37 | 41.17 | 1737.3 | 223 | 3.2 | 0.383 | 2186 | TG |
| 7 | Siziwangqi | 112.1533 | 42.0780 | 1439 | 315.2 | 4 | 0.516 | 1900 | TG |
| 8 | Wulanchabu | 113.1244 | 41.0557 | 1392 | 350.1 | 4.7 | 0.512 | 2000 | TG |
| 9 | Chayouhouqi | 113.1358 | 41.5916 | 1499 | 318.8 | 4.3 | 0.430 | 2186 | TG |
| 10 | Shangdu | 113.4799 | 41.5415 | 1361 | 337.1 | 4.2 | 0.343 | 2020 | TG |
| 11 | Xianghuangqi | 113.8587 | 42.2400 | 1351 | 270.6 | 3.9 | 0.221 | 2250 | TG |
| 12 | Huade | 113.97 | 41.88 | 1483 | 311.9 | 3.2 | 0.483 | 2050 | TG |
| 13 | Zhangbei | 114.2200 | 41.3310 | 1413 | 383.7 | 3.6 | 0.450 | 1956 | TG |
| 14 | Abagaqi | 114.9481 | 44.0294 | 1153 | 238 | 1.9 | 0.291 | 1900 | TG |
| 15 | Zhengxiangbaiqi | 115.0138 | 42.2911 | 1389 | 351.2 | 2.8 | 0.274 | 1932 | TG |
| 16 | Taipusiqi | 115.2543 | 41.9875 | 1529 | 383.5 | 2.4 | 0.469 | 1879 | TG |
| 17 | Lanqi | 115.9547 | 42.6684 | 1315 | 359.6 | 2.5 | 0.313 | 1926 | SF |
| 18 | Xilinhaote | 116.2514 | 43.8036 | 1033 | 263.5 | 3 | 0.224 | 2100 | TG |
| 19 | Keqi | 117.5389 | 43.2250 | 1038 | 391.8 | 3.2 | 0.353 | 1600 | TG |
| 20 | Linxi | 118.02 | 43.6 | 923 | 369.5 | 5.2 | 0.365 | 1826 | TG |
| 21 | Chifeng | 118.9778 | 42.3060 | 568 | 370.2 | 7.8 | 0.376 | 1700 | TG |
| 22 | Naimanqi | 120.9421 | 42.9535 | 340 | 355.9 | 7.4 | 1.928 | 1979 | TG |
| 23 | Kezuohouqi | 122.4112 | 42.9017 | 251 | 414.9 | 6.8 | 0.395 | 1782 | TG |
| 24 | Liaoyuan | 124.3416 | 42.7950 | 240 | 604.2 | 7.1 | 0.338 | 1345 | FR |
| 25 | Siping | 124.5178 | 43.1757 | 243 | 622.6 | 5.8 | 0.491 | 797 | FR |
| 26 | Qingyuan | 124.9407 | 41.8513 | 682 | 780.8 | 6.2 | 0.538 | 1033 | FR |

Deserts: DS, Desert grasslands: DG, Typical grasslands: TG, Sparse forest: SF, Forests: FR.








**TABLE 2** Species richness (No./sample area) or percentages for woody plants and herbs as well as abundance
(individual/m$^2$) in five ecosystem types

| Ecosystem types | Sites | Woody richness/percentage | Herbaceous richness/percentage | Abundance | Canopy coverages (%) |
|---|---|---|---|---|---|
| Forests | Qingyuan | 11±2/40.0±4.5 | 16±1/60.0±4.5 | 30±9 | 80-90 |
| Sparse forests | Sanggendalai | 5±2/18.0±5.2 | 24±2/82.0±5.2 | 126±8 | 20-40 |
| Typical steppe | Sanggendalai | 1±1/6.67±4.44 | 19±5/93.3±4.4 | 458±54 | 5-10 |
| Desert grasslands | Erlianhaote | 2±0/17.0±4.0 | 8±1/83.0±4.0 | 23±7 | <5 |
| Desert | Ejina | 2±0/55.7±10.4 | 2±1/44.3±10.4 | 3±4 | <5 |



**TABLE 3** Seed masses, species number and proportions of 5 dispersal types in the whole study area

| Dispersal agent types | Seed mass (mg) | Species number | Proportion in the whole (%) |
|---|---|---|---|
| Wind | 2.46±6.23 | 279 | 44.86 |
| Vertebrate | 232.09 ± 823.98 | 66 | 10.61 |
| Unassisted | 7.42±12.08 | 70 | 11.25 |
| Ants | 3.56±10.03 | 195 | 31.35 |
| Adhesive | 5.07±8.12 | 12 | 1.93 |
| Total | 50.12±172.09 | 622 | 100 |











**FIGURE 1** Mechanistic frameworks of large seed formation and then community average seed mass increment
process

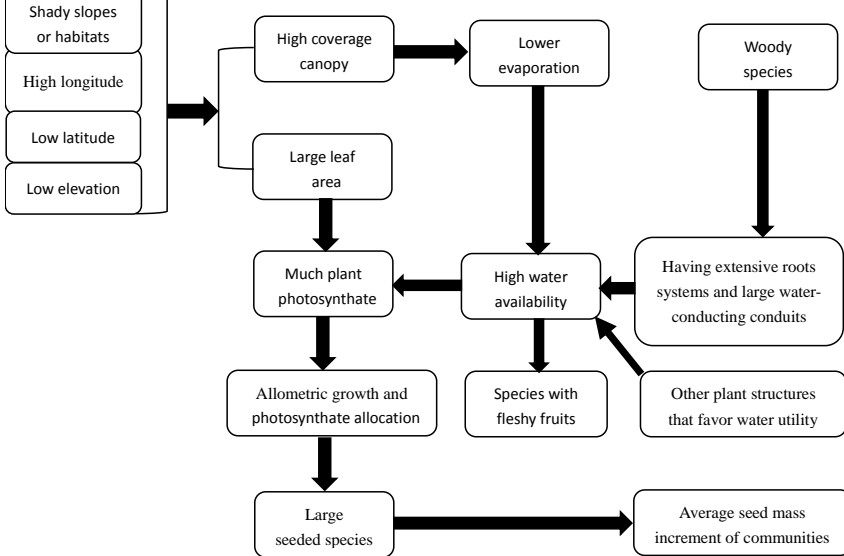













**FIGURE 2** Seed mass spectra varied among five community types in Inner Mongolia and proportions of larger seeds
and average seed mass decline from forests to desert grasslands along decreasing longitude but increase in deserts

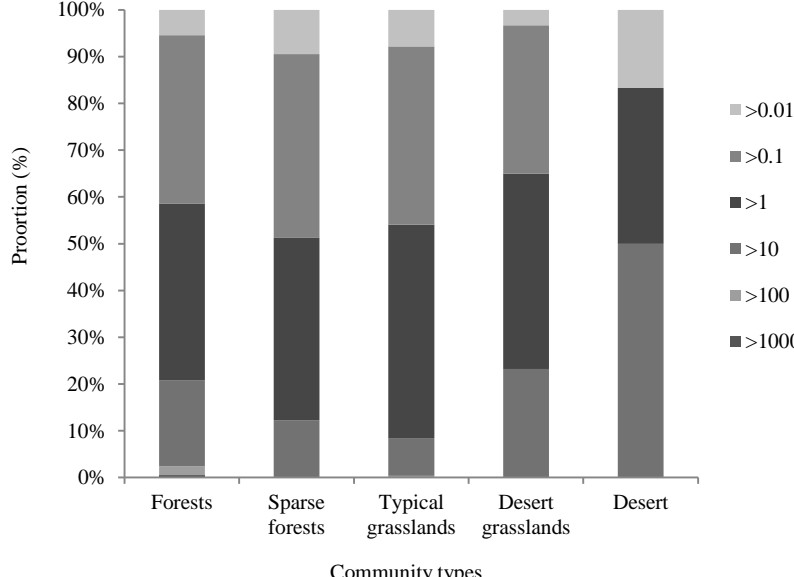


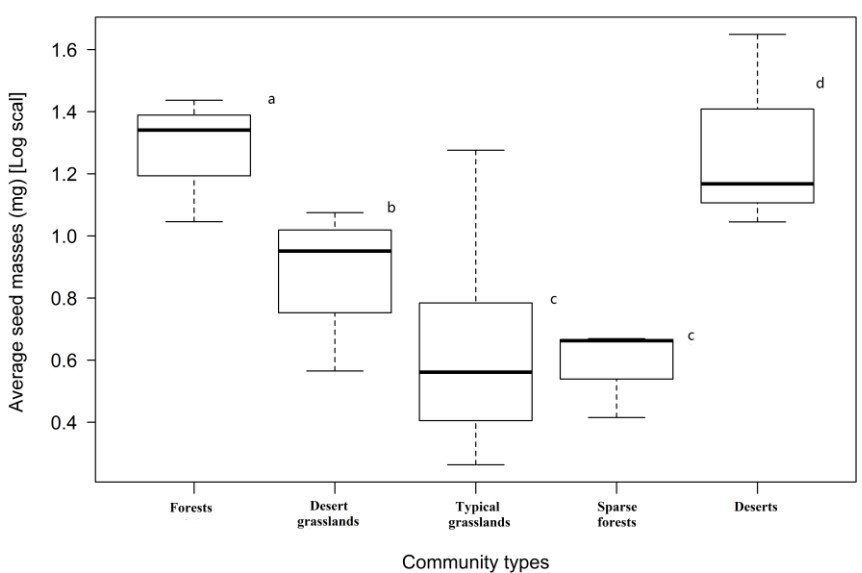





**FIGURE 3** Trees (12 species) have largest average seed mass, followed by shrubs (65 species), lianas (15 species),
subshrubs (20 species), perennial herbs (396 species) and annuals (110 species). Average seed mass of fleshy fruits is
larger than that of dry fruits in each community type

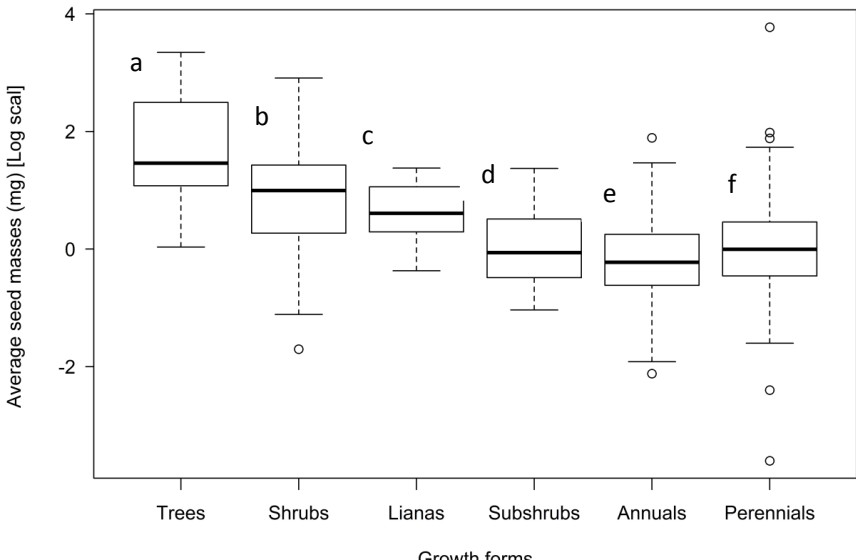


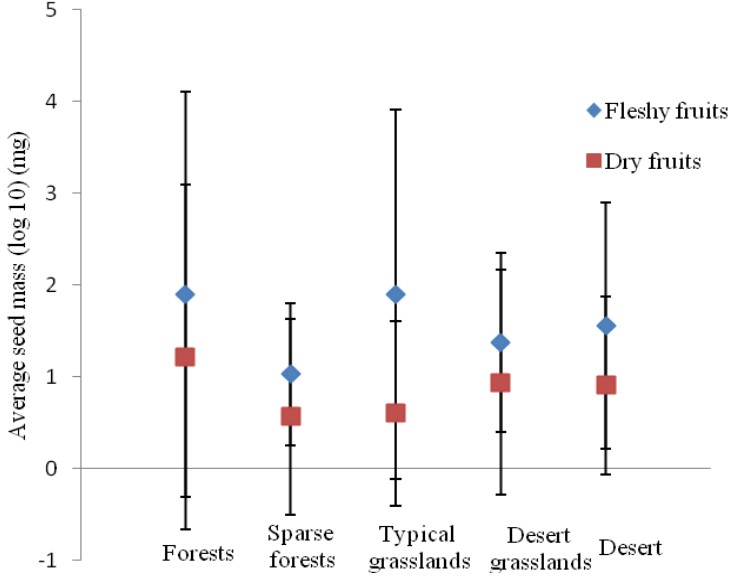




**FIGURE 4** Average seed masses of communities in 26 sampling sites (except sites of deserts) decline along rising
longitude and no significant relationships occur between seed mass and phylogenetic diversity

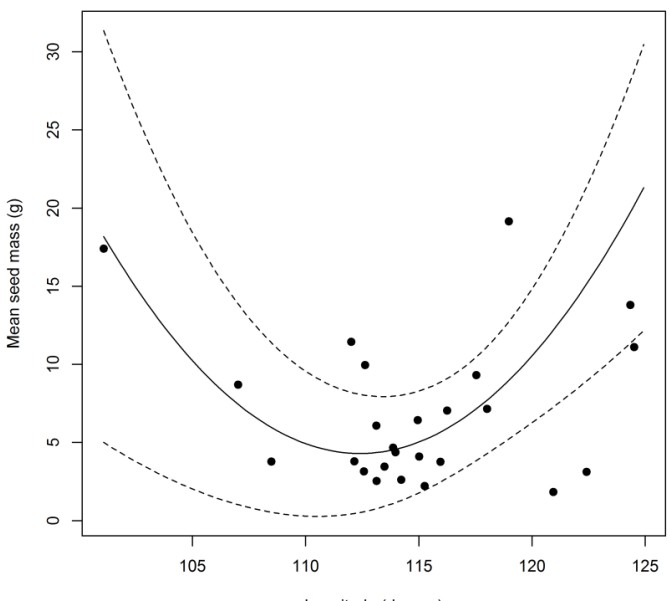


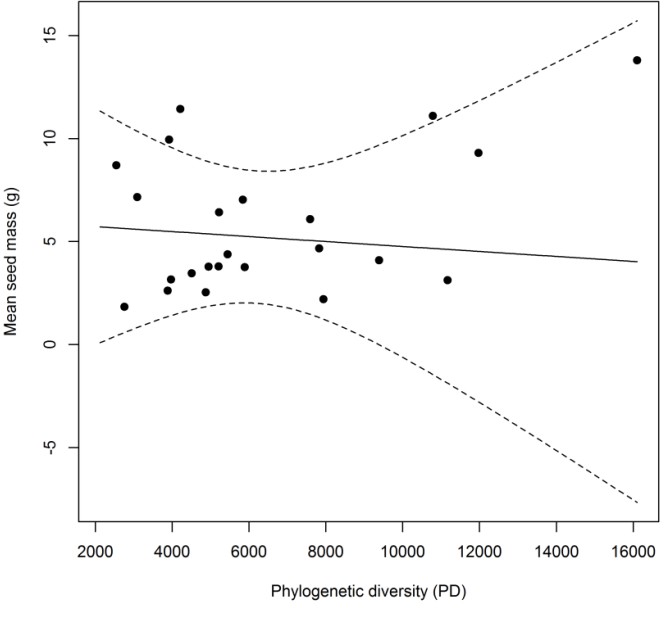




**FIGURE 5** Number of species with fleshy fruits rises in 23 sites (Ejinaqi, Wulatehouqi and Wulatezhongqi are not included) rises along increasing longitude and no significant relationships are found with phylogenetic diversity (families, genera and species)



(a)

(b)

(c)

(d)





**FIGURE 6** Proportions and species richness of plants with fleshy fruits decline gradually from forests to desert grasslands, but increase in deserts (DT: deserts, DTG: desert grasslands, TPG: Typical grassland, SFT: Sparse forests, FT: forests)

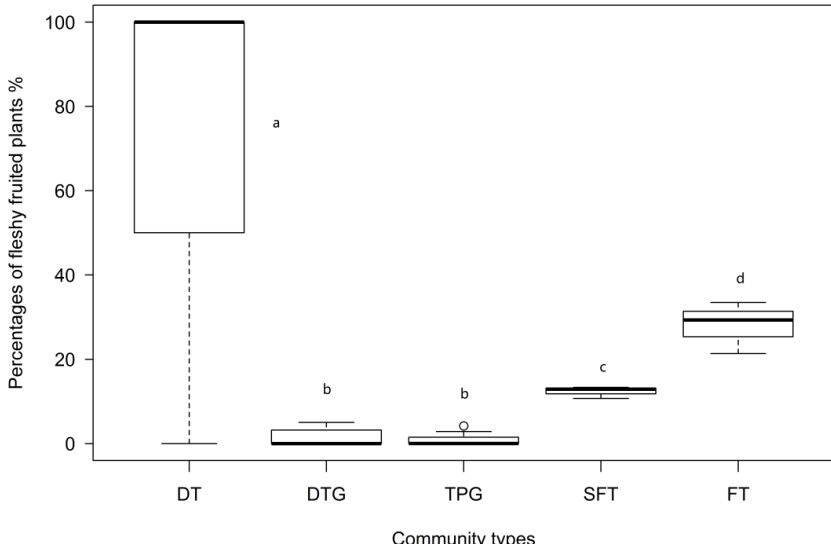

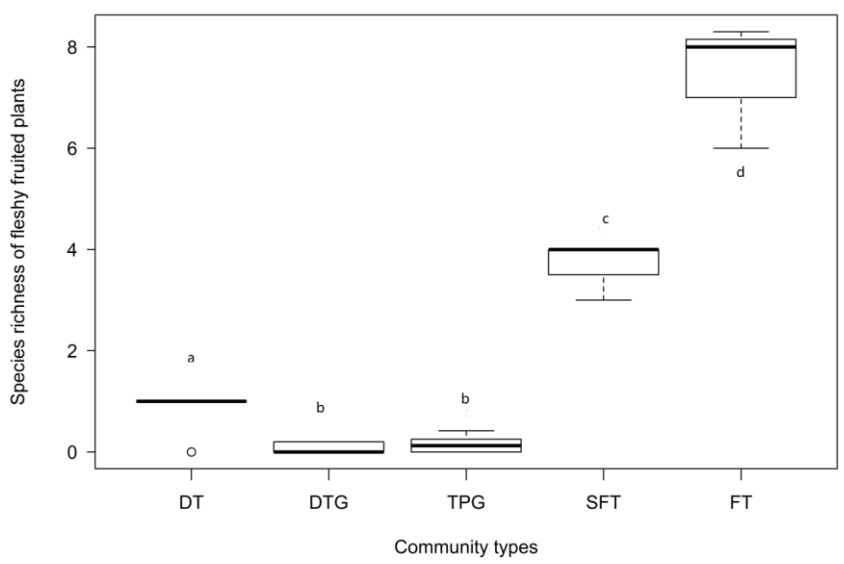