# Peer review of "Shift of seed mass and fruit type spectra along longitudinal gradient: high"

_Biogeosciences, 2019_

## Referee Comment (RC1) · Anonymous Referee #3 · 21 Jul 2020

The manuscript takes advantage of an impressive sampling of seed mass at community level across a sunshine/rainfall gradient in East Asia to address a series of correlations, including climatic, phylogenetic and functional syndromes. The authors find a decrease of seed mass and vertebrate-dispersal with decreasing rainfall among other relationships. The authors then go on to use the correlation to propose the underlying mechanisms. I am positively impressed by the amount of functional data at the community level across a series of sites along strong environmental gradients, most notably rainfall. However, the analyses can be better resolved, the manuscript structure improved, the mechanistic rationale refined and the results better presented. This means that some new analyses and extensive re-formulation is needed. Therefore, I

would propose major revision. I do not, however, expect major changes in the findings themselves, which are not necessarily ground-breaking, but I do believe are highly interesting for publishing, alone by the sheer amount of data, sampling effort and area of study. I highly encourage the authors to address the comments because these analyses will be of interest for ecologists and biogeographers in general.

Major points:

- The concept of longitudinal gradients seems to be more commonly evoke in Asia. Whereas this is ok, longitudinal gradients have limited information without further details on what environmental variables are actually being varied. Other evoked diversity gradients (latitude, elevation, depth) have clear environmental gradients (i.e. energy, temperature, light), which makes them intuitively understandable. Longitudinal gradients don't. Hence, I would rather suggest the authors to combine the name of the gradient with water availability/precipitation gradient, for example. That is more intuitive in this case. The use of the term 'longitudinal' here is important because you then control for confounding effects, such as seasonality, light/energy availability.

- L. 45-48: Note that these are not mechanisms. There are variables. If you want to mention the mechanisms (which I was hoping for), you need to mention the eco-evolutionary processes which these variables are influencing. In other words, the mechanisms are the underlying processes. I suppose you cannot state the mechanisms, as you mentioned in the sentence that follows: 'a deep understanding of the factors that underlie ... is missing'.

- L. 71-74: note that correlation is not causality. All the results are correlations, from those the authors discuss their way to propose this set of relationship as a mechanistic framework. I am ok with that, but please add in the respective discussion part that for addressing causality, you need to develop mechanistic models or lab experiments. Also, at this point, you do not need to refer to the figure, which I find more appropriate to be referred only in the end of the discussion, so it become a conclusion figure and

thus receive the last number.

- Study questions and Analyses: the authors list 4 study questions (L. 76-82), so it would be nice if the authors use this structure to present their analyses. This means, what analysis did you do to tackle each question. With this said, I find most analyses quite redundant. Why haven't the authors done a phylogenetic spatial GLMM or similar (e,g, ape package)? Plot or biome could be given as random effect to account for community assembly effects, whereas spatial models would account for spatial auto-correlation and the phylogeny for phylogenetic autocorrelation. Not sure how well the authors can fit such more complex correlative models, but there are also spatial or phylogenetic models. It just seems odd to tackle each of these correlations in separate. I generally find ok to address phylogenetic signals and I'm ok of separate tests, I'm just intrigued by the choice of models, because at least some of the analyses might be combined. Please clarify why not combining the analyses if you choose not to.

Ref: Kaldhusdal, A., Brandl, R., Müller, J., Möst, L. and Hothorn, T., 2015. Spatio‐-phylogenetic multispecies distribution models. Methods in Ecology and Evolution, 6(2), pp.187-197.

- The authors need to sort out the writing style, presentation of results, figures and tables. These are not in the same format or style.

Minor points (some are almost major though):

- Abstract, L. 20: at this point the reader does not know these five communities. So, you have to first mention the five communities. Moreover, I suppose a better term here would be biome.

- L. 26: note that this generalization rising longitude does not make sense (see major point above), as this pretty depends at what continent, part of the continent, latitude and direction you are addressing the longitude gradient.

- L. 93-94: So you have rather a sunshine and rainfall gradient that happens to be in

that particular longitudinal direction. I would ditch the longitudinal rationale and focus on the sunshine/rainfall gradient, which would make the gradient generalizable. With that said, what is the range of sunshine duration and intensity?

- L. 99: what you mean about recent?

- L. 107-109: why using just these two? Why don't you have quantification of sunshine hours or light intensity or cloud cover as well? I'm asking that because the authors mention the sunshine gradient before.

- L. 120-122: I know that you cannot re-do the sampling, which I find already very impressive. However, it is a pity that you removed wind-dispersal structures, which is part of the propagule. I suppose these would have relative low impact in the overall seed mass. Although for some small seeds, that could play a role. Can you say something about this loss of seed mass by the removal of this structures?

- L. 123-125: So what was it: based on ornamentation and appendages or based on Kew Gardens/literature? This is confusing as you provide two ways to determine dispersal mode. Please, clarify.

- L. 175-176: You can calculate MANOVA for non-independent response variables. Or the authors could be a PCA and perform regressions with the axes loadings, hypervolumes or centroids to the explanatory variables. In this way the authors would address the entire functional syndromes (i.e. trait correlations).

- L. 186-187: you can delete the half-sentence after the comma. The same in the follow-up sentence. Anyway, this entire paragraph is confusing, as you talk about average seed mass in different ways, but it reads the same.

- Tables 2-3: use the same format. Check journal formatting.

- Fig. 1: It might nicer if the authors further identify (i.e. graphically distinguish) the ecological/physiological processes (e.g. the mechanisms), related traits and the external drivers (i.e. abiotic conditions). Also, it is strange to refer to this figure before presenting the results. I see this figure more like a conclusion figure than a result figure, in which the rationale would be explained over the course of the discussion.

- Fig. 2: please, add lettering to the panels. In the top panel, add title to the legend and the units to the numbers (or the legend title). Explain the lettering in the boxplots in the figure caption. Add the statistics as well (I suppose the tests provided in the main text).

- Fig. 3: add letters to the panels and explain statistics in the caption.

- Fig. 4: the same

- Fig. 5: why were those sites excluded? And please check journal style for lettering the panels.

- Fig. 6: Use the same names and ordering to the communities as previous figures.

- L. 204: use italic font for 'p' (check throughout, there are other occasions in which the p is not in italic font)

- L. 226-227: do not use parentheses side by side, just open and close parentheses once then. This happens elsewhere as well, so check throughout.

- L. 228: check paragraph indentation

- L. 263: respectively should go inside the parentheses with the p values.

- L. 269-270: "however, mean seed masses increased from typical grasslands to desert grasslands and desert ecosystems and then to forests (Figure 2)" => This is just repeating the first half-sentence. Actually, this entire sentence is ocnfusing, because you start the sentence mentioning the longitudinal gradient of Fig. 4, but using the community types whereas the reader does not have a clue the ordination of community types along the gradient and in Fig. 4 whatsoever. Please use in the rationales only thoughts the reader can follow.

- L. 271-272: you cannot draw mechanistic explanations from correlations simply like

this. You have to explain how you think MAP and MAT might be affecting these trends, based on the indication that they might play a role due to significant relationships. I think this is a general comment that apply to other reasoning in the text (using correlations as causation). Please check throughout.

- L. 275: Note that vegetation syndromes involve trait values, you just listed trait names, not the values (e.g. the combination of low seed mass and of low fruit water content would be a syndrome).

- L. 290: larger leaves

- L. 294-295: so can you argue that the plants invest can invest more in (seedling) survival than in competitive strength?

- L. 302: can you back this up with references or with more details on this wide range?

- L. 382-283: please sort out the font sites and style. It seems it make more sense to start with the equation for St, then for Sa.

- L. 384: Sa is the average seed mass per species taken from the community total (St/n).

- L. 388: the authors should better connect this text with Fig. 1. For example, place the same symbols as parameters of the respective processes in Fig. 1.

- L. 399-401: So these are the key gradients, not longitude.

- L. 402-404: this can become its own sentence.

- L. 406-407: it is odd to say we need further studies when you just did this. Exclude this sentence, or make it more detailed about what else can be tested.

---

## Referee Comment (RC2) · Anonymous Referee #4 · 30 Sep 2020

**General comments**

In this work, the authors present a study of the variations in seed mass and other seed parameters as a function of geographic gradients in inner Mongolia and northeastern China, at the community level. The authors collected a unique dataset along many (>20) sites. These data are presented and summarized and mechanistic hypothesis of underlying processes are suggested.

I appreciated reviewing this work: the paper is interesting and well written, the effort invested in data recollection is commendable and the data collected have to potential to inform theoretical models coupling seeds features with environmental variables. Below

I provide different remarks on the text, as it can be sometimes vague and could be condensed to make the main ideas of the article more understandable and digestible by the readership.

The critical aspects concern

1. the use that is done of the data and the gap between the mechanisms involved and the methodology used

2. the fact that several times the text makes mention of a variable X strongly associated to Y, but refers to a figure that does not directly explicit this link

3. text mentioning the wrong figure

4. a discussion that ends being a bit lost in generalities and does not properly highlight the value of this work

5. generally speaking, it is difficult to define the limits of this work: what was done before the study, why this work was necessary, what has be done and what can be said with the data at hand, and what remains to be done.

**Specific comments**

1. The main point of this article seems to be to describe and summarize a number of seeds parameters (mass, phylum etc) and their relationship with geographic (longitude, latitude) and environmental (temperature, precipitation) variables.

   The nature of the work seems to be in nature more descriptive than functional, what is perfectly normal and expected, as the community needs data sets that are well presented and summarized to fit appropriate models. However, the authors tend to draw hypothesis and conclusions that can be quite remote from the data and methods used here, and sometimes subject to interpretation. With no

particularly elaborate inferential framework and no genetic data to aliment these models, it is expected that one may not always be able to link some variable to a distant mechanism. However, the connection between involved mechanism suggested by the authors (eg., selection) and the data (no genetic data) is extremely lose, making it difficult for the reader to trust the statement or to understand the limits and merits of the work (eg., l.371: "*we suggested that transition between dry fruits and fleshy fruits in response to environmental variations may also be genetically simple, involving suppression and re-expression of only a few genes*" or, l.376 "*This proves that the environment affects seed mass in the community context independent of phylogenetic constraint*"). In the absence of formal modeling or inferential framework, making such statement is indefensible.

2. A map of the area representing the main habitats and the sampling sites would complement the Table 1 fairly well, as it would allow the part of the readership that is not familiar with the biogeography of China to have a better sense of the geographic scales and ecological transitions underlying this work.

3. Reading the introduction, l.72 "*this article presents a novel mechanistic framework that integrates previous theory and hypotheses (related to climate, phylogeny, water conduction systems and other traits related to water balance) to evaluate seed mass variation among species or communities (Figure 1)*": it is unclear what is novel in this mechanistic framework, please precise. Also, this is the only time that this figure is referred (or at least correctly, see following remark) in the whole text. If this framework is worth mentioning in the abstract, we would expect further mention in the manuscript.

4. The article makes numerous general statements about the effects of "*decreasing*", "*declining*", "*increasing*" latitudes or longitudes on the flora. For example l.25: "*Due to greater water availability and increasing leaf area, much more photosynthate and allometric growth then ultimately increase the community average*

*seed mass along rising longitude (or declining latitude or elevation)*". Such statements are announced in a very general context, but are actually limited to the area of study as many areas of the world have ecological transitions happening in the reverse longitudinal trend. Please make sure that the context of the area of study is made clear. I personally found expressions such as "*from east to west*" (l.94) more intuitive.

5. Some terminology may be unclear for people foreign to the field interested by this work (eg, a mathematician, statistician or computer scientist interested by your model/data). Helping them understanding the interest of this work could be done simply by having a box briefly explaining terms like Âń *growth form*Âż , "*allometry growth theory*", "*photosynthate*" (for this last term, a brief theory is given way too late, by the end of the discussion, l.365).

6. Too many figures were not correctly referred in the text (eg. l.183). Please check that each reference to a Figure number is actually linked to the correct figure.

7. l.113-114. Mixing seeds together is a loss of data, and I would actually be curious to know how the seeds traits change according to the mother individuals too. Ideally, we want the sampled seeds traits to be independently and identically distributed variables for a same location. That is, we hope that the variation between mother plants at a same location does not overwhelm the variation between sites. Having access to the distribution of seed traits for each mother plant at each location may have enabled interesting insight on this level of variation, and does not seem too complicated to implement (if seeds are harvested directly on the mother plant) and to test statistically.

8. Table 1: Please add in the legend the complete names and/or a brief descriptive of the variables MAP, MAT, K-value, evaporation and vegetation types, so the table can be self-explanatory

9. Table 2 :Reading this table is rather difficult. Maybe the readability could be enhanced by splitting the woody and the herbaceous columns into two sub-column, rather than separating the variables richness and percentage by a slash bar.

10. Figure 3 : Please split this figure into two sub-figures (eg, 3A and 3B) for future references. The bottom figure could be made less ambiguous by slightly spacing the fleshy fruits and dry fruits bars so they don't overlap. More generally, the clarity of the manuscript could be enhanced by providing adequate labeling of Figure AND sub-figures.

11. l.380 Linking data to theory through a formal model is always useful and welcome and appreciable in biology. Here the authors provide an explicit model linking average seed mass variation to biological parameters in the discussion, but this model does not take any part in the general scientific method. Details and comments about the model are rather sparse. It is unclear how much related to the results this theory is, or how useful it is in explaining the data at hand, or what data is missing for this model to be useful. I would suggest to rewrite this paragraph. The easy way would be to remove this part, as it does not help the user understanding the interest of the work. That could be detrimental if this model has a real interest for this kind of work, or could be an easy extension of the work. In that case I would advise to provide a more ample description of the theory : how the model relates to the work presented here, how the data presented in this work could be used to inform the model, and why this has not be done, what remains to be done for this model to be useful, and references to adequate literature around this theory. l.391 "*strange patterns*" is a rather . . . strange expression for a scientific paper ;) Please replace this expression and provide a clearer explanation about what part of your results are surprising and why, and how they could have been affected by the heterogeneous distribution of groundwater in desertic sampling areas and what could be done to solve this problem. More generally, this whole paragraph sounds a bit blurry and does not promote

the quality of the discussion or the interest of the work. We advise the authors to rewrite this paragraph, with a clear statement of what could have affected the quality/results of the work, in what aspects and to what extend, what could be done to remove these limitations, and what would be too difficult/expensive to implement. l.402. I would end the sentence just before "however,".

12. The conclusion needs rewriting. In the first sentence ("*Mean seed mass, seed dispersal spectra, fruit type spectra and plant growth forms of five community types vary significantly along a longitudinal gradient, with the lowest average seed mass and the smallest proportion of species dispersed by vertebrates occurring at the middle longitude (typical grasslands)*",l.397), it is difficult to understand if the authors are making a general statement, or are describing the patterns observed in their dataset. Please clarify. The second sentence is a long list of general factors at the end of which one may wonder what factor was left out and why. It does not make a good job at summarizing the thoughts the authors have about their work, or at conveying larger implications of the study, or placing the study within the context of past research. The last sentences are very arid, and deserve more explanations (eg., what are the "*important implications in understanding origin and evolution of species with large seeds or fleshy fruits*" ?, l.405).

**Technical corrections**

- l.20 The variations of average seed mass display high congruent with transition of growth forms : this sentence seems incorrect.

- l.39 relating to plant habits do you mean habitats ?

- l.61 "Average seed mass is expected to decrease with declining longitude . . . to desert ecosystems" : this sentence does not make sense at a global scale, and seems to hold only for some regions, please precise.

- l.98 "were selected at random" : at random is not statistically rigorous, even if widely used in biological fields. You maybe mean "sampled uniformly at random" ?

- l.106 please provide adequate citation for the worldclim database and the raster package.

- l.126 "the dispersal mode represents seeds from ..." a word seems to be missing ?

- l.278 "display" you mean "displays" ?

- l.269 Please chose to address the variable "mean seed mass" as singular (mass) or plural (masses) and make it consistent along the text.

- l.272 : "see results" : please name the specific tables or figures to consult, and explicit better the sentence "MAT and MAP may be responsible ..."

- l.274 : I had to read the cited article abstract (Moles et al 2014) to understand why you cited it. Please provide a more explicit explanation on how your findings contrast the results found by Moles 2014.

- l.306. The authors mention Âń typical grasslands and desert grasslands Âż and refer to Figure 4, but it seems a mistake, as Figure 6 seems a better fit. Please go through each Figure refered in the article and make sure that you refer to the right figures and tables.

- l.309. Are the authors citing Figure 6 in the article of Yu et al, 2017 ? The typing does not seem correct, I would rather say "see Figure 6 in Yu et al. 2017" or "Zu et al, 2017, Fig. 6 ". If the authors use latex, you can use brackets to include words before and/or after a citation : something like citep[see eg,][, Fig. 6](Yu2017)

- l.310 "The increasing prevalence of fleshy-fruited species with increasing canopy coverage (Table 2)". Table 2 does not refer explicitly to fleshy fruited species, making the relationship with canopy coverage implicit. Please refer to the adequate result, or provide a better explanation, so the reader does not have to interpret what is meant. The same remark holds for l.318 and mention to Figure 3.

- l.359 Please provide citation for CO2 concentration homogeneity and small fluctuations. Same for solar radiation.

- l.360. I never heard of the term "partition out", but I'm no native speaker either. Maybe a synonym would make things clearer ?

- l. 377 independent of : independently of ?

- l. 378 : "the five communities ..." The interest of this statement is unclear. Please elaborate.

---

## Author Comment (AC3) · 22 Oct 2020

**Response to the referees**

**We thank the two reviewers for their valuable comments. According to their suggestions, we revised our article, and the revision details are as follows.**

Referee 3

1) L. 45-48: Note that these are not mechanisms. There are variables. If you want to mention the mechanisms (which I was hoping for), you need to mention the eco-evolutionary processes which these variables are influencing. In other words, the mechanisms are the underlying processes. I suppose you cannot state the mechanisms, as you mentioned in the sentence that follows: 'a deep understanding of the factors that underlie … is missing'.

**We had changed the word "mechanisms" into "ecological factors". "⋯ and several ecological factors are proposed to explain such seed mass variation gradients or patterns, for example, temperature (Moles et al., 2014),⋯⋯"**

2) L. 71-74: note that correlation is not causality. All the results are correlations, from those the authors discuss their way to propose this set of relationship as a mechanistic framework. I am ok with that, but please add in the respective discussion part that for addressing causality, you need to develop mechanistic models or lab experiments. Also, at this point, you do not need to refer to the figure, which I find more appropriate to be referred only in the end of the discussion, so it become a conclusion figure and thus receive the last number.

**We had adjusted the position of the mechanistic framework from the beginning (Introduction) to the discussion of the article so it can become a conclusion figure. We had changed FIGURE 1 into FIGURE 6 and put it in the end of the whole figures (This suggestion is very interesting. In our first edition, the mechanistic framework is at the end of the article. In order to fit the referees 2's suggestions, we revised the sentence "we construct a general hypothesis for seed mass evolution based on our conclusions and previous results" and emphasized past work basis).**

3) Study questions and Analyses: the authors list 4 study questions (L. 76-82), so it would be nice if the authors use this structure to present their analyses. This means, what analysis did you do to tackle each question. With this said, I find most analyses quite redundant.

**We had used the structure of the 4 study questions to present our description and analysis in the results and the discussion, and then delete some redundant sentences. For example, in the results, we added a paragraph "3.2 Variation of species richness, growth form spectra and abundance along the longitudinal gradient⋯".**

4) Why haven't the authors done a phylogenetic spatial GLMM or similar (e,g, ape package)? Plot or biome could be given as random effect to account for community assembly effects, whereas spatial models would account for spatial autocorrelation and the phylogeny for phylogenetic autocorrelation. Not sure how well the authors can fit such more complex correlative models, but there are also spatial or phylogenetic models. It just seems odd to tackle each of these correlations in separate. I generally find ok to address phylogenetic signals and I am ok of separate tests, I am just intrigued by the choice of models, because at least some of the analyses might be combined. Please clarify why not combining the analyses if you choose not to.

**Thank you very much for your comments. The approach you suggested is very powerful and promising. We may use it in our future work.**
**For our current work, our approach produced the expected results, although the analyses are not elegant. Combined analysis had also been conducted, but the results are not better than separate analysis we think, therefore we just provided present results. For instances, the effect of 4 variables (longitude, plant growth form, vegetation types, seed dispersal syndrome) on seed mass variation was analyzed and contribution rate of seed dispersal syndrome is the biggest in the four variables. We also analyzed the effect of 3 variables (plant growth form, vegetation types, seed dispersal syndrome) on seed mass variation and found that contribution rate of seed dispersal syndrome was the largest.**

**Minor points (some are almost major though):**
1) Abstract, L. 20: at this point the reader does not know these five communities. So, you have to first mention the five communities. Moreover, I suppose a better term here would be biome.

   **We had changed communities into biomes in the abstract. For example, in the abstract, "This study aims to explore seed mass variation patterns of different biome types along a longitudinal gradient and their underlying variation mechanisms by involving an in-depth analysis on the variation of seed mass"……**

2) L. 26: note that this generalization rising longitude does not make sense (see major point above), as this pretty depends at what continent, part of the continent, latitude and direction you are addressing the longitude gradient.

   **We had changed "along rising longitude" into "from west to east". For example, present L26 is: "Due to greater water availability and increasing leaf area, much more photosynthate (photosynthesis production) and allometric growth then ultimately increase the biome average seed mass from west to east."**

3) L. 93-94: So you have rather a sunshine and rainfall gradient that happens to be in that particular longitudinal direction. I would ditch the longitudinal rationale and focus on the sunshine/rainfall gradient, which would make the gradient generalizable. With that said, what is the range of sunshine duration and intensity?

   **We had added the range values of sunshine duration and intensity in the site description.**

Past "···due to a gradual increase in sunshine duration and intensity and decrease in rainfall (from 780.6 to 29 mm) (Table 1)" was changed in present "···due to a gradual increase in sunshine duration (3000-3200h/y) and intensity ($586 \times 10^4$ - $796 \times 10^4$ KJ/m$^2$) and decrease in rainfall (from 780.6 to 29 mm) (Table 1)."

4) L. 99: what you mean about recent?

**We had changed "recent" into "recent several". Present sentence is "Different sampling designs were used in different habitat types, owing to differences in vegetation structure and density. Within each forest plot, 6 quadrats of 10×10 m² were selected uniformly at random in undisturbed or slightly disturbed (at least in recent several years) areas."**

5) L. 107-109: why using just these two? Why don't you have quantification of sunshine hours or light intensity or cloud cover as well? I am asking that because the authors mention the sunshine gradient before.

**Precipitation and temperature have been considered to be the main ecological factors that affect plant growth in previous literature. So it is not surprise to use the two in this study. In our opinion, sunshine hours or light intensity (or cloud cover) can also affect distribution patterns of seed mass. Effect of sunshine hours or light intensity on seed mass are more complex process and they may play a certain role through rainfall and temperature. For example, sunshine hours or light intensity may play a positive role when rainfall amount is enough and on the contrary they may have a negative role when water remains shortage. In this study, from east to west, sunshine hours or light intensity is rising, however, their variation range is narrow (3000-3200h/y, $586 \times 10^4$ - $796 \times 10^4$ KJ/m$^2$). In Inner Mongolia, it is water and not sunshine that being a limit factor. So we did not analyze the two factors in detail. We had added several sentences about sunshine in the discussion.**

**The revised sentences are "Solar radiation variation is not very large along longitude (see site description) especially among typical grasslands, desert grasslands and deserts with similar elevation, therefore, its effect on seed mass variation is very small, moreover, since light is not a limited factor for growth in northern China according to our observation. Variation trend of sunshine hours or light intensity are contrary to that of rainfall amount along longitude. Only when water remain sufficient, strong light may favor plant growth and increase seed mass. For example, combination of much more belowground water with more sunshine hours or higher light intensity in Erjina may increase the average seed mass, and this may be responsible for larger seed mass in desert than in some sites of desert grasslands."**

6) L. 120-122: I know that you cannot re-do the sampling, which I find already very impressive. However, it is a pity that you removed wind-dispersal structures, which is part of the propagule. I suppose these would have relative low impact in the overall seed mass. Although for some small seeds, that could play a role. Can you say something about this loss of seed mass by the removal of this structures?

**We think that loss of seed mass overall did not affect the patterns owing to the removal of**

**wind-dispersal structures, which just is small part of the propagule as you mentioned. In previous documents, wind-dispersal structures of seeds are often removed when measuring seed mass. Since majority of the species are from Asteraceae, their propogules practically are fruits and not seeds. The removal of wind-dispersal structures may be favor selection to the results of this article.**

7) L. 123-125: So what was it: based on ornamentation and appendages or based on Kew Gardens/literature? This is confusing as you provide two ways to determine dispersal mode. Please, clarify.

**We had revised the sentence into "The dispersal modes of each species were confirmed by referring the Kew Gardens (Howe and Smallwood, 1982) and literature collection from northwest China (Liu et al., 2014)".**

8) L. 175-176: You can calculate MANOVA for non-independent response variables. Or the authors could be a PCA and perform regressions with the axes loadings, hypervolumes or centroids to the explanatory variables. In this way the authors would address the entire functional syndromes (i.e. trait correlations).

**Yes, MANOVA is more powerful. But our approach produced expected results although the analyses are little awkward.**

9) - L. 186-187: you can delete the half-sentence after the comma. The same in the follow-up sentence. Anyway, this entire paragraph is confusing, as you talk about average seed mass in different ways, but it reads the same.

**We had deleted the half-sentence after the comma in L.186-187. The present sentences are "There were considerable differences in average seed mass and seed spectra among the five community types (Figure 1). Forests have the largest average seed mass (23.45±18.34 mg) and both typical grasslands (4.75±3.93 mg) and sparse forests (4.45 ± 1.18 mg) have the lowest average seed mass."**

10) Tables 2-3: use the same format. Check journal formatting.

**We had revised format of tables and figures by journal formatting. For example, present TABLE 3 is as following**

**TABLE 3** Seed mass, species number and proportions of 5 dispersal types in the whole study area

| Dispersal agent types | Seed mass (mg) | Species number | Proportion in the whole (%) |
| --- | --- | --- | --- |
| Wind | 2.46±6.23 | 279 | 44.86 |
| Vertebrate | 232.09 ± 823.98 | 66 | 10.61 |
| Unassisted | 7.42±12.08 | 70 | 11.25 |

| | | | |
|---|---|---|---|
| Ants | 3.56±10.03 | 195 | 31.35 |
| Adhesive | 5.07±8.12 | 12 | 1.93 |
| Total | 50.12±172.09 | 622 | 100 |

11) Fig.1: It might nicer if the authors further identify (i.e. graphically distinguish) the ecological/physiological processes (e.g. the mechanisms), related traits and the external drivers (i.e. abiotic conditions). Also, it is strange to refer to this figure before presenting the results. I see this figure more like a conclusion figure than a result figure, in which the rationale would be explained over the course of the discussion.

**We had revised the frame figure and adjusted its position according to this suggestions. Present Fig.6 (past Fig.1) is as following:**

**FIGURE 6** Mechanistic frameworks of large seed formation and then community average seed mass increment process

[Figure]

12) Fig. 2: please, add lettering to the panels. In the top panel, add title to the legend and the units to the numbers (or the legend title). Explain the lettering in the boxplots in the figure caption. Add the statistics as well (I suppose the tests provided in the main text).

**We had added letters to the panels and explained the lettering in the boxplots in the figure caption. For upper figure of Fig.2 (present Fig.1), there is not bar, they are percentages of species in total species of each biomes. Present Fig.1 is as following.**

**FIGURE 1** Seed mass spectra varied among five community types in Inner Mongolia and proportions of larger seeds and average seed mass decline from forests to desert grasslands along decreasing longitude but increase in deserts (Average seed mass bearing the same letter are not significantly different at $p < 0.05$)

[Figure]

[Figure]

13) - Fig. 3: add letters to the panels and explain statistics in the caption.

**We had added letters to the panels and explained the lettering in the boxplots in the figure caption (present Fig.2). Present Fig.3 is as following.**

**FIGURE 2** Trees (12 species) have largest average seed mass, followed by shrubs (65 species), lianas (15 species), subshrubs (20 species), perennial herbs (396 species) and annuals (110 species) (2A) (Average seed mass bearing the different letter are significantly different at $p < 0.05$). Average seed mass of fleshy fruits is larger than that of dry fruits in each community type (2B) (f: fleshy fruits, d: dry fruits)

[Figure]

[Figure]

14) - Fig. 4: the same

**We had revised it (present Fig.3 is as following).**

**FIGURE 3** Relationships between average seed mass of communities and longitude and phylogenetic diversity. Average seed mass declines as longitude rises and it reaches its bottom at around 114 degrees, and after that it increases. But average seed mass do not have significant relationships with phylogenetic diversity ($p>0.05$)

[Figure]

(a)

(b)

15) - Fig. 5: why were those sites excluded? And please check journal style for lettering the panels.

**We had added results about those sites included. Some sites such as Erjina may be a special regions because of a river flowing through it. We just want to check how the results changed when exclude it.**

**We had revised Fig.4 (past Fig.5). We had corrected those mistakes about site number Present Fig.4 is as following:**

**FIGURE 4** Relationships between number of species with fleshy fruits and longitude (A, B) and phylogenetic diversity (C, D). Number of species with fleshy fruits increases as longitude increases. But it does not have significant relationship with phylogenetic diversity ($p>0.05$)

[Figure]

16) - Fig. 6: Use the same names and ordering to the communities as previous figures.

**We had revised them (adding A and B in the caption and the figure) and present Fig.5 is as following.**

**FIGURE 5** Proportions (A) and species richness (B) of plants with fleshy fruits decline gradually from forests through sparse forests to (typical and desert) grasslands, but increase in deserts (The same letter indicates difference is insignificant at $p < 0.05$)

[Figure]

(A)

[Figure]

(B)

17) - L. 204: use italic font for 'p' (check throughout, there are other occasions in which the p is not in italic font)

**We had changed into italic font for 'p'. For example, present L.204-207 is as following:**
**Seeds that are dispersed by vertebrates (232.09 ± 823.98mg) were significantly larger than those dispersed by wind (2.46±6.23 mg) (F = 238.2, *p* < 0.0001), ants (3.56 ± 10.03 mg) (F = 17.73, *p* < 0.0001), and those with unassisted dispersal (7.42 ± 12.08 mg, F=17.73, *p*=0.000) and adhesive dispersal (5.07 ± 8.12 mg, F = 17.73, *p* < 0.0001) (Table 3).**

18) - L. 226-227: do not use parentheses side by side, just open and close parentheses once then. This happens elsewhere as well, so check throughout.

**We had revised them. Present** L. **226-227 is as following.**

**···average seed mass had significantly positive relationship with longitude ($R^2 = 0.232$, $p = 0.012$) and MAP ($R^2 = 0.48$, $p = 0.00015$). Average seed mass was found to just be···**

19) - L. 228: check paragraph indentation

  **We had revised it.**

20) - L. 263: respectively should go inside the parentheses with the p values.

  **We had finished it. Present sentence is "However, relationships between number of genera or genetic diversity and longitude are not significant (respectively, $p = 0.056$ and $p = 0.058$) (Figure 4)".**

21) - L. 269-270: "however, mean seed masses increased from typical grasslands to desert grasslands and desert ecosystems and then to forests (Figure 2)" => This is just repeating the first half-sentence. Actually, this entire sentence is confusing, because you start the sentence mentioning the longitudinal gradient of Fig. 4, but using the community types whereas the reader does not have a clue the ordination of community types along the gradient and in Fig. 4 whatsoever. Please use in the rationales only thoughts the reader can follow.

  **We had deleted the repeating sentence. Present sentence is "The average seed mass displays a significantly declining trend along decreasing longitude from forests to typical grasslands and then to some sites in desert grasslands in this region (Figure 3), showing congruent distribution patterns to plant growth form spectra variation (Table 2)".**

22) - L. 271-272: you cannot draw mechanistic explanations from correlations simply like this. You have to explain how you think MAP and MAT might be affecting these trends, based on the indication that they might play a role due to significant relationships. I think this is a general comment that apply to other reasoning in the text (using correlations as causation). Please check throughout.

  **We had deleted the sentence about "mechanistic explanations" and try to explain how MAP and MAT affected these trends. In other parts of this article, we had also revised those unfit reasoning.**

23)  - L. 275: Note that vegetation syndromes involve trait values, you just listed trait names, not the values (e.g. the combination of low seed mass and of low fruit water content would be a syndrome).

  **We had revised them. Present L.275 is "The combined effects of precipitation and temperature may be, to some extent, most important to certain vegetation syndromes such as high seed mass and high fruit water content (Moles et al., 2014)."**

24) - L. 290: larger leaves

**We had revised it. Present L.290 is "Surely, woody species, on average, having larger leaves, can produce more photosynthate to invest in seeds (Díaz et al., 2016)."**

25) - L. 294-295: so can you argue that the plants invest can invest more in (seedling) survival than in competitive strength?

**We had rewrote the sentences (that make the plants invest more in propagules than in their survival and competitive strength).**

26) - L. 302: can you back this up with references or with more details on this wide range?

**We had added more details. Present sentences is "Biotic dispersal agents exert a strong selective pressure on angiosperm species with various seed size in Inner Mongolian plateau, as evidenced by the evolution of a wide range of adaptations for animal (such as ants, birds, squirrels ) dispersal".**

27) - L. 382-383: please sort out the font sites and style. It seems it make more sense to start with the equation for St, then for Sa.

**We had revised this paragraph. Present sentences are "$S_t$ is the total seed mass of all species in a community, $S_a$ is the average seed mass per species taken from the total community (St/n), *n* is number of species in a community, $C_{i1}$ is the allometric growth coefficient (or allocation portion to seeds)⋯".**

28) - L. 384: Sa is the average seed mass per species taken from the community total (St/n).

**We had corrected this sentence and please see above 27).**

29) - L. 388: the authors should better connect this text with Fig. 1. For example, place the same symbols as parameters of the respective processes in Fig. 1(present Fig.6).

**We had added some symbols in Fig.6. Please see above 11).**

30) L. 399-401: So these are the key gradients, not longitude.

**We agree with you. We had added other two environmental factors: sunshine hours or light intensity.**

31) L. 402-404: this can become its own sentence.

**We had rewrote the conclusions. The conclusions are as following.**
**"Mean seed mass, seed dispersal spectra, fruit type spectra and plant growth form spectra of**

**five biome types vary significantly along a longitudinal gradient, with the lowest average seed mass and the smallest proportion of species dispersed by vertebrates occurring at the middle longitude (typical grasslands). The selection for these propagule attributes is most likely to be driven by external and internal drivers (Figure 6), however, water availability potentials and growth-allometry may be key drivers of seed-mass variation along climatic gradients or resource gradients. Larger seeded species or species with fleshy fruits may have evolved due to much photosynthate or high water availability in plants. Our findings can provide help in understanding origin and evolution of species with large seeds or fleshy fruits."**

32) L. 406-407: it is odd to say we need further studies when you just did this. Exclude this sentence, or make it more detailed about what else can be tested.

**We had deleted the sentence "Further studies are needed to better understand the···".**

Referee 4
The critical aspects concern

1) the use that is done of the data and the gap between the mechanisms involved and the methodology used

   **We had deleted some unreasonable sentences and rewrote some sentences in order to repair the gap. For example, we had deleted "we suggested that transition between dry fruits and fleshy fruits in response to environmental variations …"(see following details).**

1) the fact that several times the text makes mention of a variable X strongly associated to Y, but refers to a figure that does not directly explicit this link

   **In this results, we surely makes mention of a variable X strongly associated to Y, but this is not a figure that does not directly explicit this link. Because those are the results from part of the research sites and not the analysis on the data of the whole research sites. Our purpose is to explore more distribution patterns of seed mass and their mechanisms. The journal do not allow to publish more figures. In fact, the 6 figures had presented main results.**

2) 3. text mentioning the wrong figure

   **We had checked twice and some similar mistakes were corrected.**

3) a discussion that ends being a bit lost in gener` alities and does not properly highlight the value of this work

   **We had revised them, rewrote some sentences and deleted some sentences. Then it may be properly highlight the value of this work now.**

4) generally speaking, it is difficult to define the limits of this work: what was done before the study, why this work was necessary, what has be done and what can be said with the data at hand, and what remains to be done.

**We had tried to revise them. For average seed mass variation, there may have different patterns in different regions with longitudinal gradient. Therefore, we startup this study. Although many works had been conducted in seed mass variation, their mechanism remains controversial and unclear. In this study, we try to find seed mass variation patterns in the regions and try to explain their reasons. Combined previous results, we provided a mechanistic frame that may be useful for future related works.**

**Specific comments**

1) The main point of this article seems to be to describe and summarize a number of seeds parameters (mass, phylum etc) and their relationship with geographic (longitude, latitude) and environmental (temperature, precipitation) variables. The nature of the work seems to be in nature more descriptive than functional, what is perfectly normal and expected, as the community needs data sets that are well presented and summarized to fit appropriate models. However, the authors tend to draw hypothesis and conclusions that can be quite remote from the data and methods used here, and sometimes subject to interpretation. With no particularly elaborate inferential framework and no genetic data to aliment these models, it is expected that one may not always be able to link some variable to a distant mechanism. However, the connection between involved mechanism suggested by the authors (eg., selection) and the data (no genetic data) is extremely lose, making it difficult for the reader to trust the statement or to understand the limits and merits of the work (eg., l.371: "we suggested that transition between dry fruits and fleshy fruits in response to environmental variations may also be genetically simple, involving suppression and re-expression of only a few genes" or, l.376 "This proves that the environment affects seed mass in the community context independent of phylogenetic constraint"). In the absence of formal modeling or inferential framework, making such statement is indefensible.

**We had deleted some those unfit sentences (for example, "we suggested that transition between dry fruits and fleshy fruits in response to environmental variations may also be genetically simple, involving suppression and re-expression of only a few genes") and revised those sentences (for instance, This proves that the environment affects seed mass in the community context independent of phylogenetic constraint").**

2) A map of the area representing the main habitats and the sampling sites would complement the Table 1 fairly well, as it would allow the part of the readership that is not familiar with the biogeography of China to have a better sense of the geographic scales and ecological transitions underlying this work.

**In the early edition of the article, there are 10 figures. Because of limiting space of the**

**journal, we just present 6 figures.**

3) Reading the introduction, l.72 "this article presents a novel mechanistic framework that integrates previous theory and hypotheses (related to climate, phylogeny, water conduction systems and other traits related to water balance) to evaluate seed mass variation among species or communities (Figure 1)": it is unclear what is novel in this mechanistic framework, please precise. Also, this is the only time that this figure is referred (or at least correctly, see following remark) in the whole text. If this framework is worth mentioning in the abstract, we would expect further mention in the manuscript.

**We had adjusted the position of figure 1 to the last one of the whole figures and its name became figure 6, being mentioned 3 times in the discussion.**

4) The article makes numerous general statements about the effects of "decreasing", "declining", "increasing" latitudes or longitudes on the flora. For example l.25: "Due to greater water availability and increasing leaf area, much more photosynthate and allometric growth then ultimately increase the community average seed mass along rising longitude (or declining latitude or elevation)". Such statements are announced in a very general context, but are actually limited to the area of study as many areas of the world have ecological transitions happening in the reverse longitudinal trend. Please make sure that the context of the area of study is made clear. I personally found expressions such as "from east to west" (l.94) more intuitive.

**They are very nice suggestions. We had corrected our unfit statement.**

5) Some terminology may be unclear for people foreign to the field interested by this work (eg, a mathematician, statistician or computer scientist interested by your model/data). Helping them understanding the interest of this work could be done simply by having a box briefly explaining terms like ´n growth form Â˙z , "allometry growth theory", "photosynthate" (for this last term, a brief theory is given way too late, by the end of the discussion, l.365).

**We had added their short explanations on the terminology. For instances, present sentence is "Due to greater water availability and increasing leaf area, much more photosynthate (photosynthesis production) and⋯".**

6) Too many figures were not correctly referred in the text (eg. l.183). Please check that each reference to a Figure number is actually linked to the correct figure.

**We had checked each reference to a figure number and corrected the wrong number.**

7) l.113-114. Mixing seeds together is a loss of data, and I would actually be curious to know how the seeds traits change according to the mother individuals too. Ideally, we want the sampled seeds traits to be independently and identically distributed variables for a same location. That is, we hope that the variation between mother plants at a same location does not overwhelm the variation between sites. Having access to the distribution of seed traits for each mother plant at

each location may have enabled interesting insight on this level of variation, and does not seem too complicated to implement (if seeds are harvested directly on the mother plant) and to test statistically.

**We agree with your opinions, in a smaller scale there also are heterogeneity for seed mass distribution. This is also an interesting question. However in this article it is not our study emphasis. If having another opportunity, we will expand these works.**

8) Table 1: Please add in the legend the complete names and/or a brief descriptive of the variables MAP, MAT, K-value, evaporation and vegetation types, so the table can be self-explanatory

**We had added their brief descriptive of the variables such as MAP, MAT, K-value, evaporation and vegetation types. The brief descriptive is following.**

**TABLE 1 Information geographic positions and environmental factors in 26 sampling sites in Inner Mongolia plateau and Northeastern China (MAP: mean annual precipitation, MAT: mean annual temperature, K-value: phylogenetic signal values, the small the values, the weak the signals. Evaporation: the change process of evaporating from a liquid to a vapor. Vegetation types: Deserts-DS, Desert grasslands-DG, Typical grasslands-TG, Sparse forest-SF, Forests-FR)**

9) Table 2 :Reading this table is rather difficult. Maybe the readability could be enhanced by splitting the woody and the herbaceous columns into two sub-column, rather than separating the variables richness and percentage by a slash bar.

**We had split the woody and the herbaceous columns into two sub-column.**

10) Figure 3 : Please split this figure into two sub-figures (eg, 3A and 3B) for future references. The bottom figure could be made less ambiguous by slightly spacing the fleshy fruits and dry fruits bars so they don't overlap. More generally, the clarity of the manuscript could be enhanced by providing adequate labeling of Figure AND sub-figures.

**We had added labeling (2A and 2B) in this figure.**

11) l.380 Linking data to theory through a formal model is always useful and welcome and appreciable in biology. Here the authors provide an explicit model linking average seed mass variation to biological parameters in the discussion, but this model does not take any part in the general scientific method. Details and comments about the model are rather sparse. It is unclear how much related to the results this theory is, or how useful it is in explaining the data at hand, or what data is missing for this model to be useful. I would suggest to rewrite this paragraph. The easy way would be to remove this part, as it does not help the user understanding the interest of the work. That could be detrimental if this model has a real interest for this kind of work, or could be an easy extension of the work. In that case I would advise to provide a more ample description of the theory: how the model relates to the work presented here, how the data

presented in this work could be used to inform the model, and why this has not be done, what remains to be done for this model to be useful, and references to adequate literature around this theory. l.391 "strange patterns" is a rather . . . strange expression for a scientific paper ;) Please replace this expression and provide a clearer explanation about what part of your results are surprising and why, and how they could have been affected by the heterogeneous distribution of groundwater in desertic sampling areas and what could be done to solve this problem. More generally, this whole paragraph sounds a bit blurry and does not promote the quality of the discussion or the interest of the work. We advise the authors to rewrite this paragraph, with a clear statement of what could have affected the quality/results of the work, in what aspects and to what extend, what could be done to remove these limitations, and what would be too difficult/expensive to implement. l.402. I would end the sentence just before "however,"

**We had rewrote this paragraph, with a clear statement of what could have affected the quality/results of the work,**

12) The conclusion needs rewriting. In the first sentence ("Mean seed mass, seed dispersal spectra, fruit type spectra and plant growth forms of five community types vary significantly along a longitudinal gradient, with the lowest average seed mass and the smallest proportion of species dispersed by vertebrates occurring at the middle longitude (typical grasslands)",l.397), it is difficult to understand if the authors are making a general statement, or are describing the patterns observed in their dataset. Please clarify. The second sentence is a long list of general factors at the end of which one may wonder what factor was left out and why. It does not make a good job at summarizing the thoughts the authors have about their work, or at conveying larger implications of the study, or placing the study within the context of past research. The last sentences are very arid, and deserve more explanations (eg., what are the "important implications in understanding origin and evolution of species with large seeds or fleshy fruits" ?, l.405).

**We had rewrote the conclusions. Present conclusion is "Mean seed mass, seed dispersal spectra, fruit type spectra and plant growth form spectra of five biome types vary significantly along a longitudinal gradient, with the lowest average seed mass and the smallest proportion of species dispersed by vertebrates occurring at the middle longitude (typical grasslands). The selection for these propagule attributes is most likely to be driven by external and internal drivers (Figure 6), however, water availability potentials and growth-allometry may be key drivers of seed-mass variation along climatic gradients or resource gradients. Larger seeded species or species with fleshy fruits may have evolved due to much photosynthate or high water availability in plants. Our findings can provide help in understanding origin and evolution of species with large seeds or fleshy fruits".**

Technical corrections
1) l.20 The variations of average seed mass display high congruent with transition of growth forms : this sentence seems incorrect.

**We had added "spectra" after growth form.**

2) l.39 relating to plant habits do you mean habitats ?

**We had deleted "habits" and revised this sentences. Present sentence is "Furthermore, as an important aspect in the reproductive biology of plants, seed mass is evolutionarily associated with and corresponds to other plant traits, relating to growth forms (for instances, trees, shrubs and herbs), life history (for example, annual plants or perennial plants) (Moles et al., 2005a), stature and canopy sizes…".**

3) l.61 "Average seed mass is expected to decrease with declining longitude . . . to desert ecosystems" : this sentence does not make sense at a global scale, and seems to hold only for some regions, please precise.

**In this sentence, we had added "in this region" to limit our research scale.**

4) l.98 "were selected at random" : at random is not statistically rigorous, even if widely used in biological fields. You maybe mean "sampled uniformly at random" ?

**We had revised the sentence and it became "Different sampling designs were used in different habitat types, owing to differences in vegetation structure and density. Within each forest plot, 6 quadrats of $10 \times 10 \text{ m}^2$ were selected at random in undisturbed or slightly disturbed (at least in recent several years) areas".**

5) l.106 please provide adequate citation for the worldclim database and the raster package.

**We had added citation for the worldclim database and the raster package. Present sentence is "Data of temperature and precipitation as well as other climatic factors were retrieved from the Wordclim database (http://www.worldclim.org/ version1.4) using R raster package (R Core Team, 2017)".**

6) l.126 "the dispersal mode represents seeds from ..." a word seems to be missing?

**We had added two words in this sentence and now it became "and the dispersal modes represent how seeds move from the parent plant to the soil surface".**

7) l.278 "display" you mean "displays" ?

**We had corrected it. I think that "displays" is right.**

8) l.269 Please chose to address the variable "mean seed mass" as singular (mass) or plural (masses) and make it consistent along the text.

**We had turned "seed masses" into "seed mass".**

9) l.272 : "see results" : please name the specific tables or figures to consult, and explicit better the sentence "MAT and MAP may be responsible ..."

**We had deleted the sentence and rewrote it. Present description is "In these sites, average seed mass was found to have significantly positive relationship with MAP and weakly positive relationship with MAT".**

10) l.274 : I had to read the cited article abstract (Moles et al 2014) to understand why you cited it. Please provide a more explicit explanation on how your findings contrast the results found by Moles 2014.

**We had revised the sentence. Now it became "The combined effects of precipitation and temperature may be, to some extent, most important to certain vegetation syndromes such as high seed mass and high fruit water content (Moles et al., 2014)".**

11) l.306. The authors mention ´n typical grasslands and desert grasslands Â˙z and refer to Figure 4, but it seems a mistake, as Figure 6 seems a better fit. Please go through each Figure refered in the article and make sure that you refer to the right figures and tables.

**We had corrected those mistakes as mentioned above.**

12) l.309. Are the authors citing Figure 6 in the article of Yu et al, 2017 ? The typing does not seem correct, I would rather say "see Figure 6 in Yu et al. 2017" or "Zu et al, 2017, Fig. 6 ". If the authors use latex, you can use brackets to include words before and/after a citation : something like citep[see eg,][, Fig. 6](Yu2017)

**We had corrected those inadequate citation. Present sentence is "Previous findings showed that fleshy fruited species were often associated with shaded habitats, mature forests, tropical forests, regions with lower elevations and woody life form (summarized in Yu et al., 2017), indicating high canopy coverage and low evaporation (Figure 6)".**

13) l.310 "The increasing prevalence of fleshy-fruited species with increasing canopy coverage (Table 2)". Table 2 does not refer explicitly to fleshy fruited species, making the relationship with canopy coverage implicit. Please refer to the adequate result, or provide a better explanation, so the reader does not have to interpret what is meant. The same remark holds for l.318 and mention to Figure 3.

**We had revised this sentence. It had become "The increasing prevalence of canopy coverage (Table 2, Figure 4) with increasing fleshy-fruited species is probably related to the prominence of species with larger seeds in such habitats".**

14) l.359 Please provide citation for CO2 concentration homogeneity and small fluctuations. Same for solar radiation.

**We had added citation for CO2 concentration homogeneity and small fluctuations and for solar radiation. The citation is as following.**

**Wang, G. C., Wen, Y. P., Kong, Q. X., Ren, L. X., Wang, M., L. Background concentration and its variation of CO2 over China Mainland. Chi. Sci. Bul., 47, 780-783, 2002.**

15) l.360. I never heard of the term "partition out", but I'm no native speaker either. Maybe a synonym would make things clearer?

**We had changed "partitioned out" into "excluded".**

16) l. 377 independent of : independently of ?

**We had checked "independent of" and we think it is right.**

17) l. 378 : "the five communities ..." The interest of this statement is unclear. Please elaborate.

**We had revised it. Present statement is "the five communities and found to be little involved in the relationships between seed mass and longitude, MAP and MAT".**

---

## Author Response (AR1)

**Response to the referees**

**We thank the two reviewers for their valuable comments. According to their suggestions, we revised our article, and the revision details are as follows.**

Referee 2

1)To make the novel mechanistic framework that you present more prominent from the beginning, rather than introducing it towards the end,

**We had adjusted the position of the mechanistic framework from the discussion to the beginning (introduction) of the article.**

2)…and implement a modelling approach that could disentangle these mechanistic processes rather than base results purely on correlations.

**According to referee's suggestions, 1) we revised the mechanistic framework, supplementing new information; 2) we implemented a model that disentangle how the seed mass improved with water increment in discussion. 3) Four references were added. 4) Abstract was revised again.**

3)Additionally, figure captions should be improved, indicating more clearly what is shown and why (related to which question), so that they can stand by themselves without having to read the whole results section.

**We had improved each figure caption in this article according to the referee's suggestions.**

Referee 1

**1)We revised our article again.**

**In addition, some small mistakes were corrected, for instances, fair to hair, in line 386.**

[revised manuscript text omitted]

= 0.0401, *p* = 0.6382), phylogenetic signals were not found to be related to longitude for the five community types.

**4 Discussion**

**4.1 Variation of seed mass spectra and environmental factors**

There is strong and consistent effect of community type (along a longitudinal gradient) on seed mass (Figure 1, Figure

3). The average seed mass displays a significantly declining trend along decreasing longitude from forests to typical grasslands and then to some sites in desert grasslands in this region (Figure 3)

In these sites, average seed mass was found to have significantly positive relationship with MAP

and weakly positive relationship with MAT. The combined effects of precipitation and temperature may be, to some extent, most important to certain vegetation syndromes such as high seed mass and high fruit water content (Moles et al., 2014). High water availability potentially can produce high assimilation products and high temperature (in normal range of plant growth) can increase water availability.

In this study average seed mass of each biome displays congruent distribution patterns to plant growth form spectra variation (Table 2, Figure 1). General linear models (GLMs) revealed significant relationships between seed mass and each of the variables predicted to influence the longitudinal gradient in seed mass: plant growth form (99.76%), vegetation types (99.01%) and seed dispersal syndrome (99.88%), as the each variable reflects different profiles of biome syndromes, not being independent effect factors to the seed mass. Such patterns have had previously been attributed mostly to a correspondence of seed mass to plant growth form and seed dispersal syndrome, which themselves are driven by climatic and environmental variations (Moles et al. 2005a; Moles et al. 2007). In Inner Mongolia, typical grasslands are often composed mainly of grasses (many of which are biennial and perennial) that are small-seeded (Figure 2), whereas trees and lianas that dominate forests and shrubs that dominate deserts have the largest seeds (Figure 2). Large seeds were proved to be often associated with woody growth forms (Salisbury, 1942; Baker, 1972; Silvertown, 1981; Mazer, 1989; Jurado et al., 1991; Elenius and Torstensson 1991; Leishman and Westoby, 1994; Moles et al., 2005a; Moles et al., 2005b). This pattern is often attributed to woody plants' better capability to take up (Schenk and Jackson, 2002; Li et al., 2002; Qi et al., 2019) and store resources and to buffer effects of environmental variations on seed size (Weiner, 2004; Moles et al., 2005a), or to reduced evaporation for understory species (Yu et al., 2017). Surely, woody species, on average, having larger leaves, can produce more photosynthate to invest in seeds (Díaz et al., 2016). Surely, woody species, on average, having larger leaves, can produce more photosynthate to invest in seeds (Díaz et al., 2016).

It is possible that larger seeds are more common in drought-prone habitats most likely because they allow seedlings to establish large root systems early, with a better chance of surviving drought (Baker, 1972; Salisbury, 1974). In this study, desert grassland and desert ecosystems are found to be dominated by shrubs that often possess larger seeds (Figure 2). In Inner Mongolian Plateau these species are seldom exposed to strong interspecific competition or shading that make the plants invest more in propagules than in vegetative apparatus for competitive strengthIn Inner Mongolian Plateau 
[revised manuscript text omitted]
, therefore, its effect on seed mass variation is very small, moreover, since light is not a limited factor for growth in northern China according to our observation. Variation trend of sunshine hours or light intensity are contrary to that of rainfall amount along longitude. Only when water remain sufficient, strong light may favor plant growth and increase seed mass. For example, combination of much more belowground water with more sunshine hours or higher light intensity in Erjina may increase its average seed mass, and this may be responsible for larger seed mass in desert than in some sites of desert grasslands.

 Therefore, combined with previous results of other studies, we deduce that drivers of seed mass spatial distribution patterns include temperature, rainfall, solar radiation, soil moisture and nutrients, leaf area, canopy coverage and their interactions, however, high water availability in plant body may be the most vital driving factor in shaping seed mass spatial distribution patterns.

 According to growth allometry, a fraction of photosynthate, coming from each increment of temperature, rainfall, soil moisture and nutrients, leaf area, canopy coverage, is considered to be allocated to seeds. In addition, biological structures (such as fair or waxiness on leaf to prevent water loss), that favor water retention in plant body would also be useful in increasing seed mass or fruit water content.

In order to understand variation mechanism of seed mass better, a simple mechanistic model is provided to trying explain quantitatively average (or total) seed mass variation between communities for one species as following:

$$S_t = \sum_{i=1}^{n} C_{i1} B_t \ (C_{i1} < 1), \ S_a = 1/n \sum_{i=1}^{n} C_{i1} B_t \ (C_{i1} < 1)$$

$$B_t = B_{id} + B_{i0} + B_l$$

$S_t$ is the total seed mass of all species in a community, $S_a$ is the average seed mass per species taken from the total community (St/n), $n$ is number of species in a community, $C_{i1}$ is the allometric growth coefficient (or allocation portion to seeds)

portion to seeds) that differ among species. $B_t$ is total biomass from photosynthate per species. $B_{id}$ value is the biomass of photosynthate related to water from conducting issues for one species, $B_{i0}$ is the biomass of photosynthate related to water from other approach (for instances, lessening evaporation), $B_l$ is the biomass of photosynthate related to leaf area (Figure 6). As we know, ecological factors affecting $S_t$ are numerous. $St$ will be developed according to other sufficient data basis. For instances, seed developing time, sunshine duration and intensity and belowground water may affect $B_t$, however, how to affect and what extent will be done conducted further in the future to improve and perfect $B_t$.

Generally, seed mass is quite phylogenetically conservative (Lord et al., 1995). However, in this study, phylogenetic signal is weak across the 26 sites (Table 1) and the phylogenetic five communitiessignal and are found to be little involved in the relationships between seed mass and longitude, MAP and MAT in the five biomes. This proves that the environment affects seed mass in the community context and phylogenetic constraints are not significant (Figure 3, 4). The five communities are in middle or late successional stages in which the main construction process is environmental filtering (effect) rather than competitive exclusion (Norden et al., 2012). The five communities are in middle or late successional stages in which the main construction process is environmental filtering (effect) rather than competitive exclusion (Norden et al., 2012).

In additions, in this study we just measure the soil moisture of top 10cm which mainly influence growth of herbs, but for the growth of shrubs and trees, rich soil water below the depth of 10 cm in some area of Ejinaqi also is useful. As mentioned above, combination of much more belowground water with more sunshine hours or higher light intensity in Erjina may increase seed mass and shape the present seed mass variation patterns in this region. As mentioned above, combination of much more belowground water with more sunshine hours or higher light intensity in Erjina can increase seed mass and may be the other ecological factors for the present seed mass patterns. Moreover, ecological scale and environmental heterogeneity often affects results of functional traits along biogeographical gradients, so further study may be necessary in larger scale (or large investigation area) to identify the results of this article.

**5 Conclusions**

Mean seed mass, seed dispersal spectra, fruit type spectra and plant growth form spectra of five biome types vary significantly along a longitudinal gradient, with the lowest average seed mass and the smallest proportion of species dispersed by vertebrates occurring at the middle longitude (typical grasslands). The selection for these propagule attributes is most likely to be driven by external and internal drivers (Figure 6), however, water availability potentials and growth-allometry may be key drivers of seed-mass variation along climatic gradients or resource gradients. Larger seeded species or species with fleshy fruits may have evolved due to much photosynthate or high water availability in plants. Our findings can provide help in understanding origin and evolution of species with large seeds or fleshy fruits.

~~Mean seed mass, seed dispersal spectra, fruit type spectra and plant growth form spectra of five biome types vary significantly along a longitudinal gradient, with the lowest average seed mass and the smallest proportion of species dispersed by vertebrates occurring at the middle longitude (typical grasslands). The selection for these propagule attributes is most likely to be driven by external and internal drivers (Figure 6), however, water availability potentials and growth-allometry may be key drivers of seed-mass variation along climatic gradients or resource gradients. Larger seeded species or species with fleshy fruits may have evolved due to much photosynthate and high water availability in plants. Our findings can provide help 
[revised manuscript text omitted]

| Ecosystem types | Sites | Woody species | | Herbaceous species | | Abundance | Canopy coverages |
|---|---|---|---|---|---|---|---|
| | | Richness | Percentage | Richness | Percentage | | |
| **Forests** | Qingyuan | 11±2 | 40.0±4.5 | 16±1 | 60.0±4.5 | 30±9 | 80-90 |
| **Sparse forests** | Sanggendalai | 5±2 | 18.0±5.2 | 24±2 | 82.0±5.2 | 126±8 | 20-40 |
| **Typical steppe** | Sanggendalai | 1±1 | 6.67±4.44 | 19±5 | 93.3±4.4 | 458±54 | 5-10 |
| **Desert grasslands** | Erlianhaote | 2±0 | 17.0±4.0 | 8±1 | 83.0±4.0 | 23±7 | ≤5 |
| **Desert** | Ejina | 2±0 | 55.7±10.4 | 2±1 | 44.3±10.4 | 3±4 | ≤5 |

| | | | | | | | |
|---|---|---|---|---|---|---|---|
|  |  |  |  |  |  |  |  |
|  |  |  |  |  |  |  |  |
|  |  |  |  |  |  |  |  |

**TABLE 3** Seed mass, species number and proportions of 5 dispersal types in the whole study area

| Dispersal agent types | Seed mass (mg) | Species number | Proportion in the whole (%) |
|---|---|---|---|
| **Wind** | 2.46±6.23 | 279 | 44.86 |
| **Vertebrate** | 232.09 ± 823.98 | 66 | 10.61 |
| **Unassisted** | 7.42±12.08 | 70 | 11.25 |
| **Ants** | 3.56±10.03 | 195 | 31.35 |
| **Adhesive** | 5.07±8.12 | 12 | 1.93 |
| **Total** | 50.12±172.09 | 622 | 100 |

**FIGURE 1** Seed mass spectra vary among five community types in Inner Mongolia and proportions of larger seeds (A)

and average seed mass decline from forests to desert grasslands along decreasing longitude in the region but increase in deserts (Average seed mass bearing the same letter are not significantly different at $p < 0.05$, B)

[Figure]

[Figure]

[Figure]

          (B)

[Figure]

———

**FIGURE 2** Trees (12 species) have largest average seed mass, followed by shrubs (65 species), lianas (15 species), subshrubs (20 species), perennial herbs (396 species) and annuals (110 species) (2A) (Average seed mass bearing the different letter are significantly different at $p < 0.05$). Average seed mass of fleshy fruits is larger than that of dry fruits in each community type (2B) (f: fleshy fruits, d: dry fruits)

[Figure]

[Figure]

[Figure]

**FIGURE 3** Relationships between average seed mass of communities and longitude (A, B) and phylogenetic diversity (C, D). Average seed mass declines as longitude rises and it reaches its bottom at around 114 degrees, and after that it increases. But average seed mass do not have significant relationships with phylogenetic diversity ($p>0.05$)

[Figure]

[Figure]

[Figure]

**FIGURE 4** Proportions (A) and species richness (B) of plants with fleshy fruits decline gradually from forests through sparse forests to (typical and desert) grasslands, but increase in deserts (The same letter indicates difference is insignificant at $p < 0.05$)

[Figure]

(A)

[Figure]

(B)

[Figure]

(a)    (b)

[Figure]

(c) ——————————————————————————— (d)

**FIGURE 5** Relationships between number of species with fleshy fruits and longitude (a), number of families and genera (b, c) as well as phylogenetic diversity (d). Number of species with fleshy fruits increases as longitude increases.

 But it does not have significant relationship with phylogenetic diversity ($p>0.05$)

[Figure]

(a)                               (b)

(c)                               (d)

**FIGURE 5** Proportions and species richness of plants with fleshy fruits decline gradually from forests to desert grasslands, but increase in deserts (The sane letter indicates difference is insignificant at $p < 0.05$)

[Figure]

[Figure]

[Figure]

**FIGURE 6** Mechanistic frameworks of large seeded species formation and then biome average seed mass increment process

[Figure]

[Figure]

---

## Author Response (AR3)

Dear editors,

I had made Fig.1 and Fig.5 clearer, but I don't know whether they are OK.

Shunli